# Don't Let It Fade: Preserving Edits in Diffusion Language Models via Token Timestep Allocation

**Woojin Kim**[1]    **Jaeyoung Do**[1,2,†]
AIDAS Laboratory, [1]ECE & [2]APAI, Seoul National University
{wjk9904, jaeyoung.do}@snu.ac.kr

## Abstract

While diffusion language models (DLMs) enable fine-grained refinement, their practical controllability remains fragile. We identify and formally characterize a central failure mode—*update-forgetting*—in which uniform, context-agnostic updates induce token-level fluctuations across timesteps, erasing earlier semantic edits and disrupting the cumulative refinement process, thereby degrading fluency and coherence. As this failure originates in uniform, context-agnostic updates, effective control demands *explicit token ordering*. We propose **Token Timestep Allocation** (TTA-DIFFUSION), which realizes *soft, semantic token ordering* via per-token timestep schedules: critical tokens are frozen early, while uncertain tokens receive continued refinement. This timestep-based ordering can be instantiated as either a fixed policy or an adaptive policy driven by task signals, thereby supporting a broad spectrum of refinement strategies. Because it operates purely at inference time, it applies uniformly across various DLMs and naturally extends to diverse supervision sources. Empirically, TTA-DIFFUSION improves controllability and fluency: on sentiment control, it yields $> 20\%$ higher accuracy and nearly halves perplexity using $< 1/5$ the steps; in detoxification, it lowers maximum toxicity (12.2 vs. 14.5) and perplexity (26.0 vs. 32.0). Together, these results demonstrate that softened ordering via timestep allocation is the critical lever for mitigating update-forgetting and achieving stable and controllable diffusion text generation.

## 1 Introduction

Diffusion language models [22, 8, 24, 11, 9] have emerged as a promising alternative to auto-regressive models by generating text through iterative refinement. They generate text in parallel and leverage bidirectional context for flexible revision, making them well-suited for tasks requiring fine-grained control. A widely adopted technique for guiding this refinement is classifier guidance, which injects external gradients at each step via an auxiliary classifier. This mechanism has demonstrated effectiveness in structural control [22, 16], semantic modulation [11], and alignment [47].

Despite its advantages, classifier-guided diffusion language models still face substantial challenges. One major limitation is the degraded fluency, as the parallel generation process weakens token-level dependencies and hinders coherent phrasing [23]. This is especially problematic for controllable generation tasks, which demand both attribute alignment and linguistic fluency [2, 25, 55, 22]. Moreover, existing diffusion-based control methods typically require hundreds of steps—often exceeding 200—to gradually enforce control, leading to significant computational overhead [22, 11].

We identify a core bottleneck behind these issues: a phenomenon we term **update-forgetting**, wherein classifier-guided modifications made at one timestep fail to persist in subsequent steps due to uniform, context-agnostic token updates. As illustrated in Figure 1, this disrupts the cumulative nature of

---

†: Corresponding author

39th Conference on Neural Information Processing Systems (NeurIPS 2025).

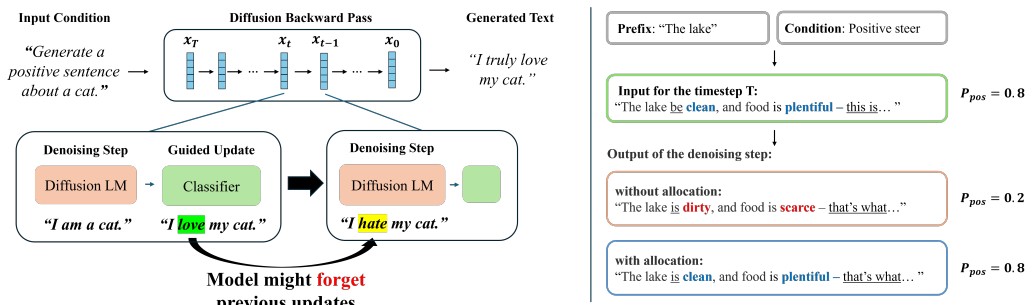

Figure 1: Illustration of *update-forgetting*. **Left:** Classifier-guided semantic edits (e.g., *love*) can be inadvertently overwritten in later denoising steps (e.g., *hate*). **Right:** This example from our experiments shows positive sentiment tokens being reversed (*clean → dirty*), undermining control; with allocation, the edits are preserved.

the iterative refinement process by erasing or overwriting previously guided changes, resulting in redundant updates, computational inefficiency, and reduced controllability.

To address this, we turn our attention to the emerging notion of *token ordering*, which determines the sequence in which tokens are updated during inference. Prior approaches have mainly explored masked diffusion, where updates are applied through unmasking strategies based on confidence [18, 34]. However, these approaches remain agnostic to semantic importance and are often limited to discrete formulations. In contrast, we propose a semantic-guided ordering mechanism that prioritizes semantically critical tokens. This design directly incorporates task-specific supervision into the update process and naturally extends to both discrete and continuous diffusion frameworks, offering a more general approach to diffusion text generation.

Building on this perspective, we introduce Token Timestep Allocation (TTA-DIFFUSION), a decoding-time framework that operationalizes token ordering through structured timestep scheduling. Instead of applying uniform, context-agnostic updates across all tokens, TTA-DIFFUSION assigns distinct timesteps that determine when and how strongly each token is refined. This enables explicit control over generation order, supporting strategies such as left-to-right decoding without requiring additional training. Beyond fixed schedules, we further propose an adaptive allocation mechanism that dynamically adjusts timesteps according to classifier gradients. By prioritizing tokens with high semantic importance, this method preserves classifier-guided edits across successive steps, mitigates update-forgetting, and ensures precise refinements on critical tokens.

We implement TTA-DIFFUSION on the simplex diffusion language model [11, 32], which facilitates seamless integration of external classifiers for guidance. To demonstrate the generality of our approach, we also validate it on continuous and discrete diffusion formulations. In addition, we further draw inspiration from progressive distillation methods [42, 3] and progressively tune the model for reduced timesteps, enabling fast generation without sacrificing control fidelity or fluency, while significantly lowering inference cost and compensating for the overhead introduced by guidance.

Consequently, TTA-DIFFUSION effectively mitigates update-forgetting by preserving semantic consistency and minimizing redundant updates throughout the generation process. Empirically, it achieves strong controllability and fluency with significantly fewer diffusion steps (as low as 100 or even 50), substantially reducing inference costs compared to prior diffusion-based methods. Beyond these performance improvements, our work positions semantic-based token ordering with inference-time timestep allocation as a principled and versatile framework for structured generation—opening new directions for efficient, fine-grained control in diffusion language models.

- We identify *update-forgetting* as a key limitation in diffusion text generation, where uniform timestep updates erase classifier-guided modifications and undermine fluency.
- We propose TTA-DIFFUSION, an inference-time framework that unifies semantic-based token ordering with structured timestep allocation to preserve guided edits and improve generation quality.
- Experimental results show that TTA-DIFFUSION outperforms baselines across tasks; in detoxification, it achieves lower maximum toxicity (*12.2 vs. 14.5*) and reduced perplexity (*26.0 vs. 32.0*) while maintaining diversity—even with under 100 timesteps.

## 2 Controllability Challenges in Diffusion Language Models

In this section, we first outline the classifier-guided update mechanism in diffusion models for language generation and then define and illustrate the *update-forgetting* phenomenon.

**Classifier-Guided Update in Diffusion** Classifier guidance in diffusion-based generation can be formulated via Bayesian inference. Given a token distribution $\tilde{x}_t \in \mathbb{R}^{N \times V}$ at timestep $t$ and a target label $y$, the goal is to maximize the posterior $P(\tilde{x}_t \mid y)$, which by Bayes' rule becomes:

$$P(\tilde{x}_t \mid y) \propto P(y \mid \tilde{x}_t)P(\tilde{x}_t).$$

Taking the gradient of the log-posterior gives:

$$\nabla_{\tilde{x}_t} \log P(\tilde{x}_t \mid y) = \nabla_{\tilde{x}_t} \log P(y \mid \tilde{x}_t) + \nabla_{\tilde{x}_t} \log P(\tilde{x}_t).$$

Since the diffusion model estimates the prior score $\nabla_{\tilde{x}_t} \log P(\tilde{x}_t)$, classifier guidance approximates the remaining term using a classifier $P_\phi(y \mid \tilde{x}_t)$:

$$\tilde{x}'_t = \tilde{x}_t + \lambda \nabla_{\tilde{x}_t} \log P_\phi(y \mid \tilde{x}_t),$$

where $\lambda$ controls the guidance strength, steering the process toward the target label.

### 2.1 Problem Definition

We identify two key instability phenomena that hinder controllability in diffusion-based text generation: *diffusion fluctuation* and *update-forgetting*. These phenomena capture how outputs may drift from their inputs or regress from previously guided states, ultimately undermining stable and effective control.

**Terminology** *Diffusion fluctuation* captures the discrepancy introduced by a single diffusion step, defined as the distance between the perturbed input and the decoded output at timestep $t + 1$. *Update-forgetting* refers to the semantic drift that occurs when guidance applied at timestep $t$ fails to persist in the next step, weakening or reversing the intended effect.

Formally, let $x_t = (x_t^1, \ldots, x_t^N)$ denote the sequence at timestep $t$, with logits $\tilde{x}_t$. The perturbed input to timestep $t + 1$ is

$$\tilde{x}_{t+1}^{\text{in}} = \tilde{x}_t + \eta_t, \quad \eta_t \sim \mathcal{N}(0, \sigma_t^2 I).$$

**Definition 1** *Diffusion Fluctuation. The fluctuation at timestep $t$ is defined as the distance between the input and the decoded output:*

$$R_t = dist(x_{t+1}, x_{t+1}^{in}),$$

*where $dist(\cdot, \cdot)$ is a general distance function.*

**Definition 2** *Update-Forgetting. The extent of update-forgetting at timestep $t$ is defined as the semantic shift between the guided sequence at $t$ and the generated sequence at $t + 1$:*

$$F_t = dist(x_t^{guided}, x_{t+1}),$$

*where $dist(\cdot, \cdot)$ measures semantic divergence.*

**Practical Deployment** In practice, the distance function $dist(\cdot, \cdot)$ can be instantiated in multiple ways. For fluctuation, we consider Hamming distance, BLEU, and BERTScore. In our analyses, we primarily adopt Hamming distance for simplicity, while also reporting BLEU and BERTScore as semantics-aware complements. For update-forgetting, we focus on the subset of tokens most influenced by guidance. Specifically, we measure the distance over the top-$k$ tokens identified as most responsive to the guidance signal, yielding a ratio that quantifies the degree to which guided effects are forgotten across timesteps.

### 2.2 Analysis

Building upon the above, we propose two hypotheses and validate them through experiments.

> **Hypothesis 1:** *Excessive fluctuation disrupts token coherence and reduces fluency.*

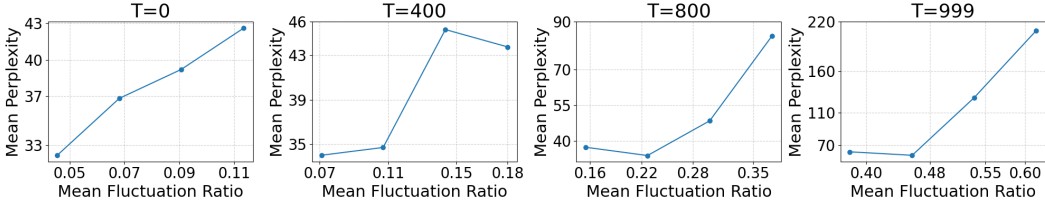

Figure 2: Fluctuation vs. perplexity across timesteps. At each timestep $t$, samples are grouped by fluctuation ratio, showing that higher fluctuation is consistently associated with higher perplexity.

**Experimental Setup.** We use the TESS [32] model with sentiment control applied via a classifier trained on the IMDB Movie Reviews dataset. Generation is performed with $T = 1000$ diffusion steps per sample under consistent decoding settings. At each intermediate timestep $t$, we compute the mean fluctuation ratio $\bar{R}_t$, defined as the average fluctuation (Normalized Hamming distance) from steps 1000 down to $t$, and record the model's perplexity at timestep $t$. For each timestep, we correlate $\bar{R}_t$ with the final perplexity across 150 generated samples to assess how fluctuation dynamics relate to fluency.

**Empirical Observation.** Figure 2 illustrates the relationship between fluctuation and perplexity. We observe a strong positive correlation, with a binned correlation of $r = 0.86$. This supports Hypothesis 1: increased fluctuation is indicative of degraded fluency. Furthermore, semantics-aware metrics show consistent trends: BLEU, BERTScore, and cosine similarity exhibit strong correlations with perplexity at timestep $t$ ($-0.6 \sim -0.8$), closely mirroring the perplexity curves. While we also observe that low-confidence tokens are more prone to fluctuation, confidence alone does not sufficiently account for the degradation in fluency; rather, excessive fluctuation emerges as the dominant factor driving reduced coherence. See Appendix B.1 for detail.

This observation is consistent with theoretical analyses of discrete diffusion models [28, 12], which establish connections between per-step transition instability and generation cross-entropy. Specifically, a higher fluctuation ratio reflects more pronounced transitions, which in turn increases the expected divergence between the model's generative distribution and the target distribution. We provide additional theoretical context in Appendix A.4.

> **Hypothesis 2:** *Update-forgetting weakens control accuracy.*

**Experimental Setup.** Using the same setup as in Hypothesis 1, we investigate how edits to *semantically pivotal tokens* change the *semantic alignment* of the output with the target attribute. For each sequence at timestep $t$, we apply classifier guidance and then perform the next diffusion update to obtain $x_{t+1}$. We compute classifier gradients with respect to the input and identify the top-5 tokens with the highest gradient norms—denoted as key tokens $\mathcal{K}_t$. We then measure the classifier's confidence in the target label **after classifier update** and **before the next classifier update**. Our analysis focuses on cases where the majority of key tokens are modified during the update, as determined by mismatch between the updated sequence $x'_t$ and the denoised output $x_{t+1}$.

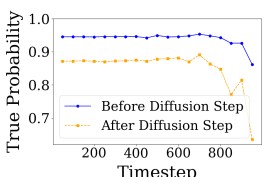

Figure 3: Classifier confidence drop due to update-forgetting.

**Empirical Observation.** Figure 3 shows a consistent drop in classifier confidence when key tokens are altered. While the average drop in probability across all steps is approximately 3–4%, the drop becomes significantly larger—often exceeding 10%—when classifier-critical tokens are modified. This provides strong evidence that update-forgetting disrupts classifier-guided control.

Based on the above observations, we substantiate the dual role of fluctuation control in classifier-guided text generation: reducing overall fluctuation enhances fluency, while stabilizing key tokens preserves classifier-driven constraints, thereby improving controllability. These findings underscore the importance of minimizing unnecessary modifications to preserve linguistic quality and improve controllable generation in diffusion language models.

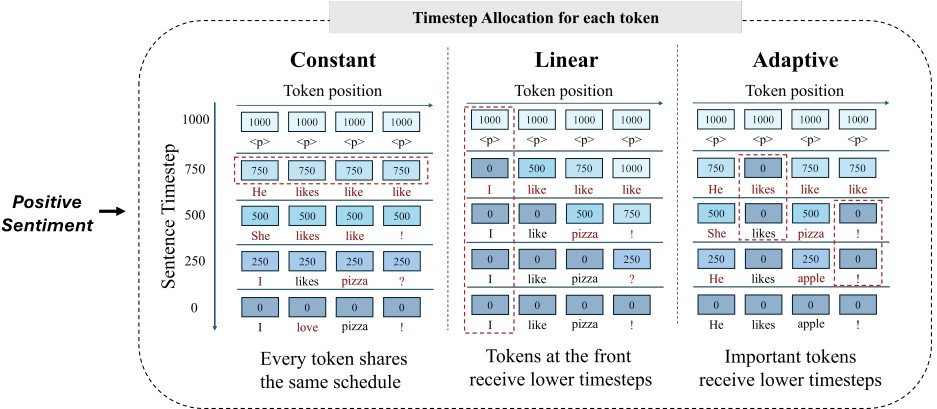

Figure 4: A comparison of token timestep allocation strategies. The left panel illustrates a default schedule in which all tokens share the same timestep. The middle panel depicts a linear schedule where timesteps decrease uniformly across tokens, allowing gradual denoising. The right panel demonstrates the adaptive schedule, in which critical tokens with high gradient values are assigned smaller timesteps, preserving important updates while refining less significant tokens.

## 3   Methodology

We begin from the perspective of **token ordering**—the order in which tokens are refined during inference. In masked diffusion frameworks, ordering has typically been implemented through heuristic unmasking strategies, such as selecting tokens based on confidence or margin [18, 34]. Once unmasked, these tokens are fixed and excluded from further refinement. While effective in certain tasks, this "hard" ordering can prematurely lock in errors and is limited to discrete formulations.

In contrast, we propose a **soft token ordering** framework that achieves ordering through timestep allocation rather than discrete masking. Instead of fixing tokens once generated, we dynamically adjust the noise schedule applied to each token, allowing them to remain revisable but with controlled update strength. This approach enables fine-grained prioritization of tokens while avoiding irreversible early commitments. Moreover, because the mechanism is based on timesteps rather than masking, it integrates naturally with not only discrete diffusion but also continuous diffusion models.

Building on this framework, we introduce **TTA-DIFFUSION** (Token Timestep Allocation), a method that dynamically assigns timesteps to tokens to enhance stability and controllability in the diffusion process. By regulating token update rates, TTA-DIFFUSION mitigates excessive fluctuations and stabilizes guided text generation. Our approach strategically prioritizes important tokens so that guided constraints persist throughout generation.

### 3.1   Token Timestep Allocation

Following AR-Diffusion [49], we assign each token $x_i$ its own timestep $t_i$. Unlike AR-Diffusion, which assigns timesteps based solely on positional movement speed in training time, we define a token timestep function that independently controls refinement for each token. Formally, let $f(i, t)$ be a token timestep function mapping the global sentence-level timestep $t$ and the token index $i$ to a local token-level timestep:

$$t_i = f(i, t).$$

A token's timestep determines its effective noise level: larger timesteps correspond to higher noise and stronger denoising, while smaller timesteps correspond to lighter refinement. By adjusting $t_i$, we can precisely control how much refinement each token undergoes, proportional to its uncertainty and contextual importance.

To formalize this, recall the forward diffusion process:

$$\tilde{x}_t \sim \mathcal{N}\left(\tilde{x}_t;\ \sqrt{\alpha_t}\,\tilde{x}_{t-1},\ (1 - \alpha_t)\,I\right),$$

where $\{\alpha_t\}_{t=1}^{T}$ is a strictly decreasing sequence of noise variance. As $t$ increases, the injected noise grows, requiring stronger denoising during the reverse process. This property enables us to tailor the refinement strength at the token level through timestep allocation (see Appendix A.6 for derivation).

Building on this principle, we design allocation schedules that distribute timesteps across a sequence of length $N$ under maximum timestep $T$. For instance, a linear schedule

$$f_{\text{linear}}(i, t) = \left\lfloor \frac{i}{N-1} t \right\rfloor$$

progressively increases timesteps with token index, keeping earlier tokens more stable while allocating greater refinement to later tokens. This process is illustrated in Figure 4.

## 3.2 Semantic-based Adaptive Allocation Strategy

While the fixed schedules in Section 3.1 enforce structured timestep assignments, they do not account for the varying importance of individual tokens. Some key tokens require minimal denoising to preserve their intended semantics, whereas others benefit from more extensive refinement. To address this limitation, we propose an *adaptive* strategy that leverages classifier gradients to guide token-wise timestep allocation.

We interpret the gradient of the classifier output with respect to each token embedding as an indicator for token importance. This intuition is supported by prior work showing that gradient-based attribution identifies important input tokens for model predictions [44, 48, 46]. In our generation process, where classifier guidance is applied between denoising steps, a high gradient magnitude indicates that a token has already been strongly shifted toward the desired attribute and is likely aligned with the control objective. To avoid overwriting this adjustment during the next denoising step, we assign a smaller timestep to such tokens. This limits further perturbation and helps preserve classifier-driven changes. We offer a simple theoretical insight into this mechanism in Appendix A.5.

Let $\{g_i\}_{i=1}^N$ be a set of gradient magnitudes derived from a task-specific classifier or scoring function, indicating the relative importance of each token. We first normalize these gradients:

$$\hat{g}_i = \frac{g_i - \min_j g_j}{\max_j g_j - \min_j g_j}, \quad i = 1, \ldots, N.$$

To adaptively allocate timesteps, we assign smaller timesteps to tokens with larger normalized gradients $\hat{g}_i$. The final timestep allocation, incorporating a smoothing factor $\alpha_{\text{smooth}}$, is defined as:

$$t_i^{\text{adaptive}} = \alpha_{\text{smooth}} t + (1 - \alpha_{\text{smooth}})(1 - \hat{g}_i) t.$$

## 3.3 Progressive Step Reduction

We build upon progressive distillation approaches [42, 3], where a student model with fewer diffusion steps is trained to imitate a teacher model trained with a larger number of steps. While effective, we found that the distillation objective becomes unstable in the final outputs, especially under small timestep regimes. Thus, we adopt a simplified training strategy that eliminates the need for distillation. Given a teacher model trained with $T$ diffusion steps, we initialize a student model and fine-tune it for $N = r \cdot T$ steps, where $r \in (0, 1]$ is the reduction ratio. Instead of matching the teacher's outputs, the student is optimized directly via cross-entropy loss over ground truth tokens $X = (x^1, \ldots, x^L)$, using its refined logits $\tilde{X}_t$ at each step:

$$\mathcal{L}_{\text{CE}} = \mathbb{E}_{X \sim \mathcal{D}} \left[ \sum_{t=1}^N \text{CE}(\text{softmax}(\tilde{X}_t), X) \right],$$

where $\tilde{X}_t$ is iteratively refined through the diffusion process.

## 4 Experiments

For our main experiments, we adopt simplex diffusion language models [11, 32] because their simplex parameterization facilitates direct utilization of external classifiers. Following the experimental setup of [11, 32], we use a pretrained RoBERTa-large [26] model, trained on a 1M subset of C4 [39], as our base model. We initially train the base model with 5000 diffusion timesteps and then progressively reduce the train steps to $\{1000, 200, 50\}$. For classifier-guided updates, we fix the guidance strength to $\lambda = 2000$. See Appendix B for further details.

Table 1: Results on detoxification and sentiment-control tasks. All models are size-matched at 330M parameters, except LD4LG (110M). For Diffusion-LM and LD4LG, maximum toxicity is not reported because the prompt is not used. For TTA-DIFFUSION, we report the variant trained with $T$=50. Results use linear allocation for toxicity and adaptive allocation for sentiment, with a smoothing factor of 0.6.

| Model | Toxicity | | | | Sentiment Control | | |
| | Avg. tox↓ | Max. tox↓ | PPL↓ | Dist-3↑ | Acc↑ | PPL↓ | Dist-3↑ |
|---|---|---|---|---|---|---|---|
| **Auto-regressive Baselines** | | | | | | | |
| PPLM | 30.6 | 59.7 | 107.4 | 0.95 | 42.6 | 201.1 | 0.94 |
| GeDi | 22.0 | 36.1 | 98.8 | **0.94** | 79.9 | 98.6 | 0.91 |
| DExperts | 15.1 | 32.0 | 48.0 | 0.87 | 83.2 | 31.8 | 0.93 |
| Air-decoding | 18.5 | 40.4 | 49.0 | 0.93 | 82.6 | 27.1 | 0.94 |
| LM-Steer | 19.1 | 47.0 | 44.4 | 0.91 | 85.4 | 78.8 | 0.86 |
| **Diffusion Baselines** | | | | | | | |
| Diffusion-LM$_{T=2000}$ | 21.8 | - | 131.2 | 0.94 | 72.8 | 89.3 | 0.94 |
| SSD-LM$_{T=1000}$ | 24.6 | 50.3 | 58.3 | 0.94 | 76.2 | 51.1 | **0.94** |
| LD4LG$_{T=250}$ | 14.5 | - | 296.4 | 0.90 | 59.9 | 70.7 | 0.95 |
| TESS$_{T=1000}$ | 14.6 | 32.3 | 58.8 | 0.92 | 71.1 | 31.7 | 0.85 |
| **Ours** | | | | | | | |
| TTA (50) $_{T=200}$ | **12.2** | **26.0** | **40.6** | 0.92 | **94.7** | **20.5** | 0.86 |
| TTA (50) $_{T=100}$ | 12.2 | 26.7 | 46.3 | 0.93 | 92.7 | 28.7 | 0.86 |
| TTA (50) $_{T=50}$ | 12.5 | 27.3 | 59.5 | **0.94** | 88.7 | 47.3 | 0.87 |

## 4.1 Tasks & Evaluation Metrics

**Detoxification.** This task aims to generate non-toxic text while preserving fluency and semantic coherence. Following Qian et al. [37], we use the challenging subset of RealToxicityPrompts [6] and select 203 prompts. For each, we generate 20 sequences with a maximum sequence length of 64.

**Sentiment Control.** This task involves generating text that adheres to sentiment constraints, such as positive or negative. Using 15 prompts from PPLM [2], we generate 50 sequences per sentiment.

**Topic Control** We generate text for four target topics—*World*, *Business*, *Sports*, and *Sci-Tech*—using 20 prompts from PPLM [2], with 20 sequences generated per topic.

**Lexically-Constrained Generation** Following Li et al. [22], we generate text under lexical constraints, including syntax trees, syntactic spans, and length constraints. Using 200 constraints applied to 50 sentences each. We train BERT-base [4] on the E2E dataset [35] to align with Li et al. [22].

**Evaluation Metrics** To evaluate overall generation quality, we measure fluency and diversity. Fluency is calculated using **Perplexity (PPL)** with GPT-2 Large [38], while diversity is evaluated using **Dist-3** [21]. For each task, we measure control strength via APIs or classifiers. For details, see Appendix B.

## 4.2 Baselines

**Auto-regressive Baselines.** **PPLM** [2] modifies the hidden representations of the model with gradient ascent. **GeDi** [19] leverages CC-LM to guide generation. **DExperts** [25] contrasts expert and anti-expert models to regulate output distribution. **Air-decoding** [55] conditions generation on prefix-based language models, while **LM-Steer** [10] adjusts word embeddings to enforce control.

**Diffusion Baselines. Diffusion-LM** [22] performs end-to-end training in embedding space, while **SSD-LM** [11] adopts a simplex-based method in vocabulary space. **LD4LG** [29] operates in latent space with control codes for guided generation. As our architecture builds on **TESS** [32], we include it for comparison.

## 4.3 Main Results

We present the results for detoxification and sentiment control in Table 1, and lexical control in Table 2. Detailed results, including those for topic control, are provided in Appendix E. Step-by-step outputs and full generation examples can be found in Appendix F.

**Detoxification and Sentiment Control**   As shown in Table 1, TTA-DIFFUSION consistently achieves the lowest toxicity in the detoxification task, demonstrating strong robustness even at a reduced timestep of $T = 50$. In terms of fluency, while TTA-DIFFUSION with $T = 200$ achieves the lowest perplexity, configurations with $T = 100$ maintain comparable levels. For sentiment control, TTA-DIFFUSION outperforms all baselines in accuracy, with the $T = 50$ setting already surpassing others, and achieving up to a 10% improvement at $T = 200$. Although perplexity increases slightly at lower timesteps (e.g., $T = 50$), it remains within an acceptable range and still compares favorably to baseline models.

**Lexically-Constrained Generation**   In lexically-constrained generation, we compare TTA-DIFFUSION with Diffusion-LM [22] at $T = 200$. As shown in Table 2, TTA-DIFFUSION significantly improves fluency over Diffusion-LM, reducing perplexity from 248.6 to 111.4 while maintaining comparable lexical accuracy.

Table 2: Lexical Control Results.  Results of Diffusion-LM are from Li et al. [22].

| Metric | Diffusion-LM | TTA-DIFFUSION |
|---|---|---|
| Syntax Tree Acc ($\uparrow$) | 86.0 | **93.1** |
| Syntax Span Acc ($\uparrow$) | **93.8** | 92.3 |
| Length Acc ($\uparrow$) | 99.9 | **100.0** |
| Mean PPL ($\downarrow$) | 248.6 | **111.4** |

## 4.4  Ablation Study

**Allocation for Continuous and Discrete Diffusion**   Our context-dependent timestep allocation extends beyond simplex diffusion models to both continuous and discrete formulations. In *continuous diffusion*, we apply the adaptive schedule to Diffusion-LM [22] for sentiment control and observe gains in both controllability and fluency: accuracy increases from 72.8% to 75.6%, while perplexity drops from 89.3 to 77.9. In *discrete diffusion*, we incorporate adaptive allocation into D-CBG [43] for molecular prop-

Table 3: Adaptive allocation results on discrete diffusion.

| $\gamma$ | Method | Valid (%) | Mean Property |
|---|---|---|---|
| 1 | D-CBG | 989 | 0.474 |
| | + Adaptive | **998** | **0.494** |
| 10 | D-CBG | 721 | 0.585 |
| | + Adaptive | **756** | **0.591** |

erty maximization on QM9 by adjusting transition probabilities according to the same scoring signal used by D-CBG; as summarized in Table 3, adaptive allocation improves validity and increases the mean property score ($0.474 \rightarrow 0.494$ and $0.585 \rightarrow 0.591$). Taken together, these results clarify that adaptive allocation yields consistent, measurable benefits across diffusion regimes, not just in the simplex-based setting.

**Effect of Timestep Allocation**   Table 4a presents the impact of token-level timestep allocation on both the base model TTA (5000) and its step-reduced variant TTA (50) across detoxification and sentiment control tasks. At relatively low timesteps, introducing a scheduling strategy substantially improves both control accuracy and language fluency. For instance, at $T = 50$, scheduling reduces toxicity from 14.0 to 12.5 and improves sentiment accuracy from 83.5% to 85.9%, while simultaneously lowering perplexity. However, we observe diminishing benefits of allocation as the number of denoising steps increases. In the case of T=1000, while some improvements are visible (e.g. sentiment accuracy from 92.2 to 92.6), the relative gains are less pronounced. Detailed results across a wider range of schedules are provided in Appendix D, E.1.

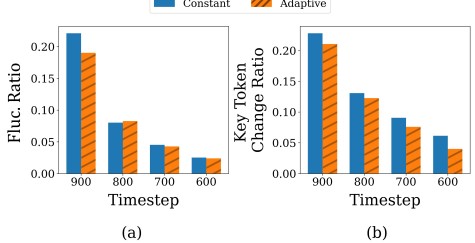

Figure 5: Comparison of constant and adaptive allocation on (a) fluctuation ratio and (b) key-token change ratio.

Also, to analyze the impact of scheduling on fluctuation and update-forgetting, we compare an adaptive allocation with a constant allocation by evaluating the fluctuation ratio and key-token change ratio ($k = 5$) for the sentiment control task. As shown in Figure 5, the adaptive allocation consistently reduces fluctuation and mitigates update-forgetting.

**Transferability of Timestep Allocation**   While developed for classifier-guided control, our timestep allocation readily extends to general generation tasks with classifier feedback. We demonstrate its effectiveness on paraphrasing and long-form generation. For paraphrasing, we follow Yuan et al. [51] using the QQP dataset with a timestep of 50. Token importance is estimated via paraphrase

Table 4: Results of scheduling and timestep allocation on generation tasks.

(a) Detoxification and sentiment control.

| Model | T | Detoxification | | Sentiment | |
|---|---|---|---|---|---|
| | | Tox. ↓ | PPL ↓ | Acc. ↑ | PPL ↓ |
| TTA (5000) | 200 | 13.2 | 630.4 | 80.8 | 47.3 |
| + with schedule | | **12.8** | **70.8** | **82.1** | **35.5** |
| TTA (50) | 50 | 14.0 | 68.0 | 83.5 | 44.0 |
| + with schedule | | **12.5** | **59.5** | **85.9** | **40.2** |

(b) Paraphrase generation.

| Model | T | BLEU ↑ | BERT ↑ | R-L ↑ |
|---|---|---|---|---|
| TTA (5000) | 1000 | 28.9 | 84.3 | 59.8 |
| TTA (1000) | 1000 | 29.8 | 85.3 | 60.7 |
| TTA (200) | 200 | 30.1 | 85.2 | 60.9 |
| + with schedule | 200 | **30.3** | **85.5** | 61.3 |
| + with schedule | 30 | **30.3** | **85.5** | **61.5** |

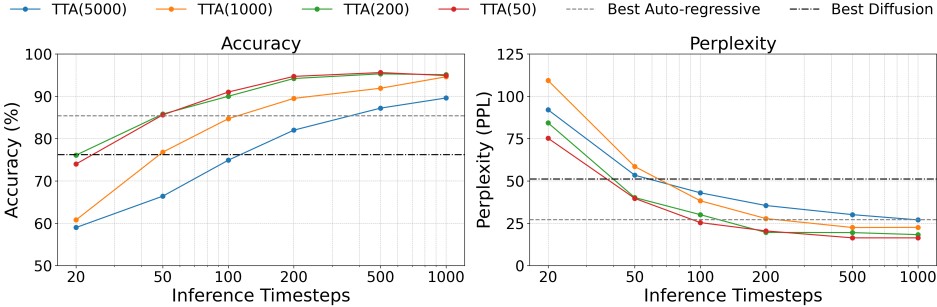

Figure 6: Sentiment control performance of models progressively fine-tuned with $t$ diffusion steps (TTA($t$)). Left: sentiment accuracy (%). Right: perplexity (PPL) across inference steps. Dashed lines denote the best-performing autoregressive (gray) and diffusion (black) baselines.

classifier. Table 4b shows improved results with allocation compared to a uniform baseline. In long-form generation (up to 512 tokens), we reuse prompts from sentiment control experiment and apply a fluency classifier with adaptive allocation at timestep 50. While unguided generation leads to extreme perplexity, guidance with allocation stabilizes it around 25–30. Further details are provided in Appendix B.2.

**About Progressive Step Reduction** Figure 6 presents the results of progressive step reduction on the sentiment control task. As shown in the figure, while all models converge to similar performance under large inference budgets (e.g., $T = 1000$), significant differences emerge under low timesteps. Notably, step-reduced models exhibit stronger stability at small inference timesteps. For perplexity, although the original $T = 5000$ model achieves competitive results only at high inference timesteps (around $T = 1000$), the progressively fine-tuned models attain comparable perplexity with as few as $T = 100$ inference steps, highlighting the efficiency gained through step reduction. See Appendix A.6, Table 9, for the detailed speedup comparison.

**Effect of Different Allocation Strategies** We evaluate the accuracy and fluency of a sentiment control task under several timestep allocation strategies. To isolate the effect of timestep reduction, we compare perplexity against the base TTA(5000) target (rather than TTA(50) reported in the main results). The random schedule samples a timestep for each token uniformly from $(0, T)$, where $T$ is the maximum timestep. The fixed schedule uses a predetermined timestep, either $0$ or $T$.

As shown in Figure 7, the *adaptive* schedule consistently achieves the lowest perplexity (best fluency). *linear* and *backward-linear* schedules also reduce perplexity relative to a constant schedule, but their gains are setting-dependent (with backward-linear sometimes edging linear). By contrast, *random* and *fixed* allocations are suboptimal; in particular, fixed-at-$T$ exhibits large variance with occasional extreme values.

# 5 Related Works

**Diffusion Language Models** The application of diffusion models to text generation has garnered increasing interest [1, 22, 8, 24, 12, 28]. A key challenge in this area is bridging the gap between discrete text and the continuous diffusion framework, which has been approached through embedding-based methods [22, 24, 8, 49], latent modeling [29, 53], and discrete mapping [13, 54, 40, 34, 41].

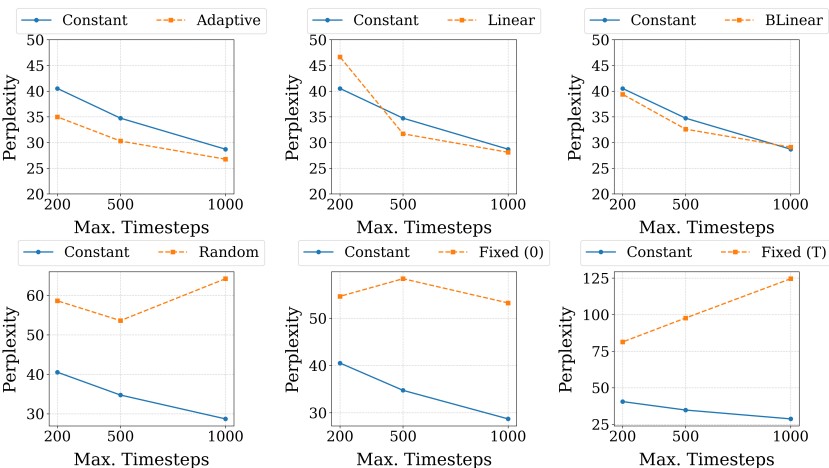

Figure 7: Effect of different token allocation strategies on perplexity. We compare linear, backward linear (BLinear), random, and fixed schedules (at timesteps 0 and T), as well as the adaptive schedule, against a constant allocation baseline (no scheduling).

Recent studies have explored simplex mapping to operate in vocabulary space [11, 32, 47]. For controllable text generation, the iterative nature of diffusion models enhances control over the generation process [22, 11, 29, 27, 16, 30, 43]. However, applying identical noise to all tokens disrupts local dependencies and harms coherence. Prior efforts on token-level noise modeling [51] and multi-level diffusion [49] primarily modify training objectives and thus offer limited leverage at inference time. We instead introduce an inference-time adaptive allocation strategy that schedules denoising timesteps token-wise to improve controllability. This perspective connects naturally to the emerging notion of *token ordering*—the sequence in which tokens are (un)masked and revised—studied in discrete diffusion [18, 34]; our approach realizes a *soft* ordering via timestep scheduling, making it applicable beyond discrete unmasking schemes.

**Controllable Text Generation**    As language models scale, effective mechanisms for controlling their outputs become essential. Training-based approaches [17, 56, 36] demand extensive computational resources and large-scale annotated datasets, constraining their scalability. In contrast, inference-stage methods offer a more flexible solution by enforcing constraints through contrastive techniques [19, 25], prompt tuning [20, 50], or modulation via latent space [5, 10]. However, these methods offer only indirect control and remain constrained by the model's inherent generative capabilities. Plug-and-play methods [2, 33], which employ external classifiers to steer generation, are closely related to our work. Despite their effectiveness, these methods struggle in auto-regressive models due to sequential dependencies that hinder the modification of previously generated tokens.

## 6    Conclusion

In this work, we identify *update-forgetting* as a central limitation in controllable text generation with diffusion language models, wherein uniform token updates overwrite previously guided modifications and disrupt the generation trajectory, compromising both fluency and controllability. To address this, we propose TTA-DIFFUSION, an inference-time framework that implements *soft token ordering* by dynamically allocating denoising timesteps across tokens according to their semantic importance. By prioritizing critical tokens and structuring the update schedule accordingly, TTA-DIFFUSION improves generation stability and enhances controllability. Our findings demonstrate that token-wise timestep allocation provides a principled mechanism for structured generation. TTA-DIFFUSION not only improves quality and efficiency through progressive reduction but also introduces a flexible, inference-time approach to operationalize token ordering without additional training. While this work adopts classifier-informed allocation, future research may explore alternative heuristics, semantic-aware scheduling, or extensions to multi-attribute control. More broadly, we view TTA-DIFFUSION as a step toward general-purpose frameworks that integrate soft ordering strategies into diffusion models, opening promising directions for efficient, fine-grained, and controllable text generation.

**Acknowledgements**

This work was supported in part by National Research Foundation of Korea (NRF) grant (RS-2025-00560762, RS-2024-00414981), and Institute of Information & communications Technology Planning & Evaluation (IITP) grant (RS-2025-02263754, RS-2025-25442338, IITP-2025-RS-2024-00397085, RS-2021-II211343). J. Do is with ASRI, Seoul National University.

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

# Supplementary Material

## Contents

# A Diffusion Formulation

## A.1 Diffusion Model Formulation

Diffusion probabilistic models [14] are generative models that learn a probabilistic distribution by iteratively denoising a latent variable, following a Markov chain. The diffusion forward pass progressively corrupts the original data representation into pure noise, and a learned reverse process (backward pass) recovers the original representation by iterative denoising.

**Diffusion Forward Pass**   The forward process follows a Markov process that transforms a given data point $\tilde{x}_0$ into a sequence of latent variables $\{\tilde{x}_t\}_{t=1}^T$ by iteratively injecting Gaussian noise. This process is defined as:

$$q(\tilde{x}_t \mid \tilde{x}_{t-1}) = \mathcal{N}\big(\tilde{x}_t; \sqrt{\alpha_t}\, \tilde{x}_{t-1}, (1 - \alpha_t)I\big),$$

where $\alpha_t$ is the noise schedule that controls the scale of noise. For sufficiently large T and appropriate noise schedule, the final distribution of $\tilde{x}_T$ converges to pure Gaussian noise $\mathcal{N}(0, I)$.

**Diffusion Backward Pass**   To recover the original data from pure noise, we approximate the reverse Markov process, which progressively removes injected noise at each timestep, ultimately reconstructing $\tilde{x}_0$. Given the sampling from prior $\tilde{x}_T \sim \mathcal{N}(0, I)$, the reverse process is modeled as:

$$p_\theta(\tilde{x}_{t-1} \mid \tilde{x}_t) \approx \mathcal{N}\big(\tilde{x}_{t-1}; \mu_\theta(\tilde{x}_t, t), \Sigma_\theta(\tilde{x}_t, t)\big),$$

where $\mu_\theta(\tilde{x}_t, t)$ and $\Sigma_\theta(\tilde{x}_t, t)$ are parameterized by a neural network trained to predict the denoised representation at timestep $t - 1$.

**Guided Update Procedure**   To incorporate classifier control within the generation procedure, the diffusion backward process can be modified using an external classifier $p(y \mid \tilde{x}_t)$.

Controlling $\tilde{x}_0, \dots, \tilde{x}_T$ corresponds to sampling from the posterior:

$$p(\tilde{x}_0, \dots, \tilde{x}_T \mid y) = \prod_{t=1}^T p(\tilde{x}_{t-1} \mid \tilde{x}_t, y).$$

By applying Bayes' rule, the conditional transition at each step is expressed as:

$$p(\tilde{x}_{t-1} \mid \tilde{x}_t, y) \propto p(\tilde{x}_{t-1} \mid \tilde{x}_t) \cdot p(y \mid \tilde{x}_{t-1}, \tilde{x}_t).$$

Following prior work on controlled diffusion models [22], we simplify the dependency structure using a conditional independence assumption.

$$p(y \mid \tilde{x}_{t-1}, \tilde{x}_t) = p(y \mid \tilde{x}_{t-1}).$$

Under this assumption, the gradient-based update at each step of the reverse process is given by:

$$\nabla_{\tilde{x}_{t-1}} \log p(\tilde{x}_{t-1} \mid \tilde{x}_t, y) =$$
$$\nabla_{\tilde{x}_{t-1}} \log p(\tilde{x}_{t-1} \mid \tilde{x}_t) + \nabla_{\tilde{x}_{t-1}} \log p(y \mid \tilde{x}_{t-1}).$$

where $\log p(\tilde{x}_{t-1} \mid \tilde{x}_t)$ is modeled by the diffusion process and $\log p(y \mid \tilde{x}_{t-1})$ is modeled by an external classifier.

## A.2 Simplex Diffusion Language Model Formulation

This section provides a detailed formulation of the simplex diffusion process applied to diffusion language modeling.

**Simplex Mapping**   Since text is inherently discrete, we map tokens into a continuous simplex space following [11]. Let $\mathcal{V}$ be the vocabulary set, and $x \in \mathcal{V}$ be a token. The token-to-simplex transformation is defined as:

$$\tilde{x}^i = \begin{cases} +K, & \text{if } i = \text{index}(x) \\ -K, & \text{otherwise} \end{cases}$$

where $K$ is a predefined simplex constant. This mapping produces a logit representation $\tilde{X}_0$ over the vocabulary, forming the input for the diffusion process.

**Forward Diffusion Process**   The forward process for standard diffusion process described above now becomes:

$$q(\tilde{X}_t \mid \tilde{X}_{t-1}) = \mathcal{N}\big(\tilde{X}_t; \sqrt{\alpha_t}\tilde{X}_{t-1}, (1 - \alpha_t)I\big),$$

The latent state at any step $t$ can be expressed directly as:

$$\tilde{X}_t = \sqrt{\bar{\alpha}_t}\tilde{X}_0 + \sqrt{1 - \bar{\alpha}_t}Z, \quad Z \sim \mathcal{N}(0, K^2 I).$$

**Backward Diffusion Process**   To reconstruct the original representation, we approximate the reverse diffusion process as:

$$p_\theta(\tilde{X}_{t-1} \mid \tilde{X}_t) \approx \mathcal{N}\big(\tilde{X}_{t-1}; \mu_\theta(\tilde{X}_t, t), \Sigma_\theta(\tilde{X}_t, t)\big),$$

where $\mu_\theta(\tilde{X}_t, t)$ and $\Sigma_\theta(\tilde{X}_t, t)$ are predicted by a neural network trained to estimate the denoised representation.

**Sampling and Decoding**   Sampling begins from a Gaussian prior $\tilde{X}_T \sim \mathcal{N}(0, K^2 I)$ and iteratively refines through reverse diffusion:

$$\tilde{X}_{t-1} = \sqrt{\bar{\alpha}_{t-1}}\hat{X}_t + \sqrt{1 - \bar{\alpha}_{t-1}}Z,$$

where $Z \sim \mathcal{N}(0, K^2 I)$. At each step, logits are predicted from the current noisy simplex representation:

$$X_{\text{logits},t} = \text{logits}_\theta(\tilde{X}_t, t).$$

The corresponding token distribution is obtained via softmax:

$$P(X_t) = \text{softmax}(X_{\text{logits},t}).$$

To project back into the vocabulary space, we employ top-$p$ sampling [15] as in [11], ensuring diversity while preserving fluency. The selected token is mapped back to the simplex representation as:

$$\tilde{X}_t^i = \begin{cases} +K, & \text{if } i = \text{top-}p(P(X_t)) \\ -K, & \text{otherwise.} \end{cases}$$

[11] noted that this transformation maintains the sampled representation near the original simplex structure while ensuring the model retains its non-autoregressive properties

## A.3   Connection of Simplex Diffusion to Uniform–State Discrete Diffusion

Let $V = |\mathcal{V}|$ and let $e_y \in \{0, 1\}^V$ be the one–hot vector for token $y$. Our simplex embedding uses the constant $K > 0$ and maps $y$ to

$$\psi(y) := K(2e_y - \mathbf{1}) \in \mathbb{R}^V,$$

so the correct coordinate is $+K$ and all others are $-K$. The forward (continuous) corruption used in §A.2 is

$$\tilde{X}_t \overset{d}{=} \sqrt{\bar{\alpha}_t}\,\psi(y) + \sqrt{1 - \bar{\alpha}_t}\,Z, \qquad Z \sim \mathcal{N}(0, K^2 I_V). \tag{1}$$

Define the discrete projection $z_t := \arg\max(\tilde{X}_t) \in \{e_1, \ldots, e_V\}$.

**Normalization to the Gaussian dual form.**   There exist scalars $a_t > 0$ and $b_t \in \mathbb{R}$ such that the affine reparameterization

$$w_t := a_t\,\tilde{X}_t + b_t\,\mathbf{1}$$

obeys

$$w_t \overset{d}{=} \mathcal{N}\big(\tilde{\alpha}_t\,e_y,\ (1 - \tilde{\alpha}_t^2)\,I_V\big), \tag{2}$$

with the (Gaussian) correlation parameter

$$\tilde{\alpha}_t^2 = \frac{4\bar{\alpha}_t}{1 + 3\bar{\alpha}_t}, \qquad a_t = \frac{1}{K\sqrt{1 + 3\bar{\alpha}_t}}, \qquad b_t = \frac{\sqrt{\bar{\alpha}_t}}{\sqrt{1 + 3\bar{\alpha}_t}}. \tag{3}$$

Moreover, $\arg\max(w_t) = \arg\max(\tilde{X}_t)$ because $\arg\max$ is invariant to adding $b_t \mathbf{1}$ and to positive rescaling $a_t$.

**Discrete marginals via Diffusion Duality.** Applying the Diffusion Duality result of Sahoo et al. [41] to (2), the pushforward of $w_t$ (equivalently, of $\tilde{X}_t$) under $\arg\max$ yields the marginals of a Uniform–State Discrete Diffusion Model (USDM):

$$\mathbb{P}(z_t = \cdot \mid z_0 = e_y) = \text{Cat}\left(\cdot;\; \alpha_t^{\text{disc}} e_y + (1 - \alpha_t^{\text{disc}})\frac{1}{V}\mathbf{1}\right), \qquad \alpha_t^{\text{disc}} = T(\tilde{\alpha}_t), \quad (4)$$

where $T : [0,1] \to [0,1]$ is the diffusion transformation operator of Sahoo et al. [41] (their Eq. (10)), defined for a $V$-class simplex. Combining (3) and (4) gives the induced discrete schedule directly from the simplex process:

$$\alpha_t^{\text{disc}} = T\left(\sqrt{\frac{4\,\bar{\alpha}_t}{1 + 3\,\bar{\alpha}_t}}\right). \quad (5)$$

**Remarks.** The map $\bar{\alpha}_t \mapsto \tilde{\alpha}_t$ in (3) is monotone and *independent of the embedding scale $K$* due to the matched noise variance $Z \sim \mathcal{N}(0, K^2 I_V)$ in (1).

## A.4 Relation Between Diffusion Fluctuation and Perplexity

**Proposition.** *High empirical diffusion fluctuation, when measured as a deviation from an optimal baseline, is a theoretically grounded indicator of a high upper bound on model perplexity. This follows because excess fluctuation lower-bounds the per-step denoising error that appears in the perplexity certificate.*

**Formalism.** We adopt the discrete diffusion setting of Haxholli et al. [12]. By definition, $\text{PPL} = \exp\big(\mathcal{H}(p_0, p_\theta)\big)$, and Theorem 4 of Haxholli et al. [12] provides an *upper bound* on the cross-entropy of the form

$$\mathcal{H}(p_0, p_\theta) \leq \text{UB}(\mathcal{H}) \equiv C + \int_0^1 \mathbb{E}_{x_0, x_t}\big[D_{\text{KL}}\big(q(\cdot \mid x_t) \,\|\, p_\theta(\cdot \mid x_t)\big)\big]\, dt, \quad (6)$$

where $q(\cdot \mid x_t)$ is the Bayes-optimal reverse kernel and $p_\theta(\cdot \mid x_t)$ is the learned reverse kernel.

Let $L$ be the sequence length, $d_H(\cdot, \cdot)$ the Hamming distance, and define the (sampled) fluctuation rates at time $t$ by

$$R_t^{(p)}(x_t) := \mathbb{E}_{X \sim p_\theta(\cdot \mid x_t)}\left[\frac{1}{L}\,d_H\big(X, x_t\big)\right], \quad (7)$$

$$R_t^{(q)}(x_t) := \mathbb{E}_{Y \sim q(\cdot \mid x_t)}\left[\frac{1}{L}\,d_H\big(Y, x_t\big)\right]. \quad (8)$$

The *excess fluctuation* is $\Delta R_t(x_t) = R_t^{(p)}(x_t) - R_t^{(q)}(x_t)$.

**Proof of relation.** **Step 1: Excess fluctuation $\Rightarrow$ total variation.** For the bounded function $f(z) = \frac{1}{L}d_H(z, x_t) \in [0, 1]$,

$$\big|\Delta R_t(x_t)\big| = \big|\mathbb{E}_{p_\theta}[f] - \mathbb{E}_q[f]\big| \leq d_{\text{TV}}\big(q(\cdot \mid x_t), p_\theta(\cdot \mid x_t)\big), \quad (9)$$

which is the standard variational bound for TV when $f \in [0, 1]$.

**Step 2: TV $\Rightarrow$ KL (Pinsker).** Pinsker's inequality yields

$$d_{\text{TV}}\big(q(\cdot \mid x_t), p_\theta(\cdot \mid x_t)\big) \leq \sqrt{\tfrac{1}{2} D_{\text{KL}}\big(q(\cdot \mid x_t) \,\|\, p_\theta(\cdot \mid x_t)\big)}. \quad (10)$$

Combining (9) and (10) gives the pointwise lower bound

$$D_{\text{KL}}\big(q(\cdot \mid x_t) \,\|\, p_\theta(\cdot \mid x_t)\big) \geq 2\big(\Delta R_t(x_t)\big)^2. \quad (11)$$

**Step 3: Impact on the cross-entropy upper bound.** Taking expectations over $(x_0, x_t)$ (the forward process) and integrating over $t$ in (11),

$$\int_0^1 \mathbb{E}_{x_0, x_t}\big[D_{\text{KL}}\big(q(\cdot \mid x_t) \,\|\, p_\theta(\cdot \mid x_t)\big)\big]\, dt \geq 2\int_0^1 \mathbb{E}_{x_0, x_t}\big[\big(\Delta R_t(x_t)\big)^2\big]\, dt. \quad (12)$$

Substituting (12) into (6) yields

$$\mathrm{UB}(\mathcal{H}) \ \geq \ C \ + \ 2 \int_0^1 \mathbb{E}_{x_0, x_t}\left[\left(\Delta R_t(x_t)\right)^2\right] dt. \tag{13}$$

Therefore, the large fluctuation forces the certified upper bound to be large. Equivalently,

$$\mathrm{UB}(\mathrm{PPL}) \ = \ \exp\left(\mathrm{UB}(\mathcal{H})\right) \ \geq \ \exp\left(C + 2 \int_0^1 \mathbb{E}_{x_0, x_t}\left[\left(\Delta R_t(x_t)\right)^2\right] dt\right).$$

**Conclusion.** The excess fluctuation $\Delta R_t$ is a principled indicator of generation quality: large deviations from the Bayes-optimal flip rate imply large per-step denoising KL, which enlarges the certified upper bound on cross-entropy and thus on perplexity. Empirically observed instability is therefore a direct, theoretically grounded symptom of underlying denoising error.

### A.5   Theoretical rationale for token–timestep allocation

**Setup and notation.** Let $x = (x^1, \ldots, x^N)$ be a sequence of length $N$ and $t \in [0, T]$ the global reverse time. A *schedule* is a mapping $f : \{1, \ldots, N\} \times [0, T] \to [0, T]$ with local times $t_i = f(i, t)$. We assume $f$ is *monotone in time*: $\partial t_i / \partial t \in [\rho_{\min}, \rho_{\max}]$ with $0 < \rho_{\min} \leq \rho_{\max} < \infty$. Let $\alpha_i(t) \in (0, 1]$ be the signal coefficient, and define the per-token forward variance $\sigma_i^2(t) = 1 - \alpha_i(t)$. At reverse time $t$, we impose a *budget constraint* $\sum_{i=1}^N \sigma_i^2(t) = C(t)$ with *box constraints* $\sigma_{\min}^2(t) \leq \sigma_i^2(t) \leq \sigma_{\max}^2(t)$ induced by the integrator and training schedule. Classifier guidance augments the score by $\lambda \nabla_{\tilde{x}} \log p(y \mid \tilde{x}_t)$; write $g_i(t) := \left\|\nabla_{\tilde{x}^i} \mathcal{L}_{\mathrm{clf}}(\tilde{x}_t)\right\|$ and let $\hat{g}_i(t) \in [0, 1]$ be a normalized importance weight.

**Assumptions.** (A1) (*Score-error bound form*) The cross-entropy upper bound can be written (or upper bounded) as

$$J_2 \ \leq \ \int_0^T \sum_{i=1}^N \phi_i\left(\sigma_i^2(t)\right) dt,$$

where each $\phi_i : [0, 1] \to \mathbb{R}_{\geq 0}$ is nondecreasing and *convex*. (A2) (*Local sensitivity*) For small perturbations of the noise around a baseline, $\phi_i(\sigma_i^2) \approx a_i + b_i \sigma_i^2$ with $b_i \propto w_i$ and weights $w_i := w_i(t)$ nondecreasing in $g_i(t)$. (A3) (*Margin smoothness*) The classifier margin $m(\tilde{x})$ is $L$-smooth: $\|\nabla m(u) - \nabla m(v)\| \leq L\|u - v\|$. Under a one-step Euler–Maruyama update with independent Gaussian perturbations $\varepsilon^i \sim \mathcal{N}(0, \sigma_i^2 I)$, the expected one-step margin drop satisfies

$$\mathbb{E}[\Delta m] \ \leq \ \frac{L}{2} \sum_{i=1}^N \eta_i \, \sigma_i^2, \quad \text{with} \quad \eta_i \ \leq \ c_0 + c_1 \, g_i$$

for constants $c_0, c_1 \geq 0$ determined by the diffusion and guidance drifts.

**Perplexity bound improvement.** By (A1) and (A2), at each $t$ minimizing the instantaneous contribution to $J_2$ under the budget and box constraints reduces to

$$\min_{\{\sigma_i^2\}} \sum_{i=1}^N b_i \, \sigma_i^2 \quad \text{s.t.} \quad \sum_{i=1}^N \sigma_i^2 = C(t), \ \ \sigma_{\min}^2 \leq \sigma_i^2 \leq \sigma_{\max}^2. \tag{$\star$}$$

Because the objective is linear and the feasible set is a product of intervals intersected with a simplex, the optimum assigns *smaller* $\sigma_i^2$ to *larger* $b_i$ until box constraints are met. When $C(t)$ lies strictly inside the box (no saturation), the interior KKT solution is *affine in the weights*:

$$\sigma_i^2(t) \ = \ \Pi_{[\sigma_{\min}^2, \sigma_{\max}^2]}\left(c(t) \ - \ \kappa(t) \, b_i(t)\right)$$

for Lagrange multipliers $c(t), \kappa(t) > 0$, where $\Pi$ denotes clipping to the box. Since $b_i$ is nondecreasing in $g_i$, a practical normalized rule

$$\sigma_i^2(t) \ \propto \ 1 - \hat{g}_i(t)$$

is a realizable instance of the KKT solution up to scaling and clipping. Integrating over $t$ yields $J_2(f_{\mathrm{adaptive}}) \leq J_2(f_{\mathrm{constant}})$, with *strict* inequality whenever the $b_i$ are not all equal and no box constraint forces equality. By Haxholli et al. [12], the adaptive schedule yields a tighter upper bound, $\mathrm{UB}_{\mathrm{PPL}}(f_{\mathrm{adaptive}}) < \mathrm{UB}_{\mathrm{PPL}}(f_{\mathrm{constant}})$ whenever the above conditions hold.

**Impact on control accuracy.** By (A3), the expected one-step margin drop obeys

$$\mathbb{E}[\Delta m] \ \leq \ \tfrac{L}{2} \sum_{i=1}^{N} \eta_i \, \sigma_i^2 \quad \text{with} \quad \eta_i \text{ non-decreasing in } g_i.$$

Minimizing this upper bound under the same budget and box constraints is the same program as $(\star)$ with $b_i \leftarrow \eta_i$, hence has the same KKT form and the same monotone allocation: assign *less* noise to larger $g_i$. Therefore the adaptive rule $\sigma_i^2 \propto (1 - \hat{g}_i)$ minimizes the margin-drop bound among feasible schedules, yielding

$$\mathrm{ACC}(f_{\text{adaptive}}) \ \geq \ \mathrm{ACC}(f_{\text{constant}}),$$

with strict improvement under the same "no-saturation and non-identical weights" condition.

**Conclusion.** Under (A1)–(A3) and a fixed per-time noise budget with box constraints, the allocation $\sigma_i^2 \propto (1 - \hat{g}_i)$ is the affine KKT solution of the per-time convex programs that (i) minimize the cross-entropy upper bound integrand and (ii) minimize the one-step margin-drop bound. Hence it is a *Pareto improvement* for both perplexity (via the bound) and control accuracy. Strict improvements obtain whenever token importances are not identical and box constraints do not force equality.

## A.6 Relation of Denoising Ratio and Diffusion Timestep

We analyze the relationship between the magnitude of denoising updates and the diffusion timestep. Specifically, we show that the noise injected at timestep $t + 1$ is greater than the noise injected at timestep $t$, implying that later timesteps require proportionally larger denoising updates.

We examine the forward diffusion process given by

$$\tilde{x}_t \ = \ \sqrt{\alpha_t}\, \tilde{x}_{t-1} \ + \ \sqrt{1 - \alpha_t}\, \epsilon_t, \quad \epsilon_t \sim \mathcal{N}(0, I),$$

where $\{\alpha_t\} \subset (0, 1]$ is a strictly decreasing sequence in $t$. We show below that the variance of the newly injected noise at timestep $(t + 1)$ is strictly larger than that at timestep $t$.

First, observe that the newly injected noise at step $t$ can be written as

$$\text{Noise}_t \ = \ \sqrt{1 - \alpha_t}\, \epsilon_t,$$

where $\epsilon_t \sim \mathcal{N}(0, I)$. Its variance is then

$$\mathrm{Var}\big[\text{Noise}_t\big] \ = \ (1 - \alpha_t)\, I.$$

Next, because $\{\alpha_t\}$ is strictly decreasing,

$$\alpha_{t+1} \ < \ \alpha_t \quad \Longrightarrow \quad 1 - \alpha_{t+1} \ > \ 1 - \alpha_t.$$

Thus, for the noise term at step $(t + 1)$,

$$\begin{aligned}
\mathrm{Var}\Big(\sqrt{1 - \alpha_{t+1}}\, \epsilon_{t+1}\Big) \ &= \ (1 - \alpha_{t+1})\, I \\
&> \ (1 - \alpha_t)\, I,
\end{aligned}$$

which is exactly $\mathrm{Var}[\text{Noise}_t]$. Consequently, the noise injected at each successive timestep $(t + 1)$ is strictly larger in variance than the noise injected at timestep $t$. In summary,

$$\mathrm{Var}\Big(\sqrt{1 - \alpha_{t+1}}\, \epsilon_{t+1}\Big) \ > \ \mathrm{Var}\Big(\sqrt{1 - \alpha_t}\, \epsilon_t\Big).$$

Since the forward diffusion process injects increasing noise at later timesteps, the corresponding reverse (denoising) steps must compensate by removing larger noise magnitudes. Hence, the magnitude of the denoising update is greater for higher timesteps in the reverse diffusion chain.

.

# B  Experimental Details

## B.1  Experiment Configuration for Fluctuation and Update-Forgetting

We conduct our analysis using a TESS-based diffusion language model [32] trained with subset of C4, equivalent to a fully non-autoregressive extension of SSD-LM. For controllability, we apply classifier-guided sentiment control using a classifier trained on the IMDB Movie Reviews dataset, with a guidance strength of $\lambda = 2000$. Each sample is generated using $T = 1000$ diffusion steps under fixed decoding parameters.

**Fluctuation–Perplexity Analysis.**  To quantify how token-level fluctuations affect fluency, we compute the mean fluctuation ratio $\bar{R}_t$ from steps $T$ down to each intermediate timestep $t$ (i.e., over $\{T, T-1, \ldots, t\}$). For each timestep $t$, we correlate $\bar{R}_t$ with the final sequence perplexity across 150 samples. We repeat this procedure using BLEU and BERTScore (F1) computed between each step's output and the previous step's sequence. This allows us to track how instability during the denoising process relates to semantic drift and final output quality. To evaluate lexical subtlety, we also detect synonym replacements using WordNet. Across all timesteps, only 0.117 tokens per 64-token sequence (on average) are replaced by synonyms, suggesting that most fluctuations involve non-trivial changes.

**Update-Forgetting Analysis.**  For each sequence at timestep $t$, we compute classifier gradients and select the top-5 tokens with highest gradient norm as key tokens $\mathcal{K}_t$. We record classifier confidence before and after a single diffusion step and compare it for cases where at least 3 of the 5 key tokens are modified. The "before" state refers to the output immediately after classifier guidance, while the "after" state is the output following noise injection and denoising. The classifier's confidence is measured as the probability assigned to the target label.

**Correlation Computation.**  We compute Pearson correlation coefficients between fluctuation ratio and semantic similarity metrics (BLEU, BERTScore), as well as between those metrics and final perplexity. All correlations are computed across (sample, timestep) pairs from 150 generations. In addition, we compute correlation trends within timestep bins to isolate convergence effects in later steps. Even in the final 100 steps, we observe non-trivial correlations (e.g., BLEU–perplexity: $-0.23$), indicating that sample-level fluctuation trajectories meaningfully affect final outcomes.

**Synonym Analysis by Timestep.**  To rule out the possibility that observed fluctuations are predominantly benign, we compute the proportion of changed tokens that qualify as WordNet synonyms within 100-step bins. The proportion remains consistently low (1–5%) even in later timesteps, reinforcing the interpretation that update-forgetting and fluctuation often correspond to semantically meaningful deviations rather than minor paraphrasing.

## B.2  TTA-DIFFUSION Configuration

**RoBERTa-large**  RoBERTa-large [26] serves as the base model for sentiment/topic-controlled text generation, and detoxification tasks. For training, we extract a 1M subset from the C4 dataset[39]. The model was fine-tuned for 300K steps using a batch size of 64 and a learning rate of $3 \times 10^{-5}$. During training, we randomly select between 2 and 10 tokens as a prefix, with the model learning to generate the remaining tokens. The training objective follows the standard cross-entropy loss, as in [11, 32]. A cosine noise schedule [45] with a simplex parameter of $k = 5$ was employed. The training process used 5,000 diffusion timesteps, For progressive step reduction, we begin with a base model trained using 5000 diffusion steps and progressively reduce the number of steps to 1000, 200, and 50. The base model is trained for a maximum of 300K timesteps, while each fine-tuned model is trained until convergence (300K, 100K, 40K). Training was performed on a single NVIDIA H100 GPU, requiring approximately 50 hours to complete.

**BERT-base**  BERT-base [4] is utilized to maintain consistency in model size with [22], which employs BERT-base [4] as the base architecture. The model is trained on the E2E dataset [35] using a batch size of 50 and a learning rate of $1 \times 10^{-4}$. The maximum sequence length is set to 64, with 2,000 diffusion steps during training and 200 steps during inference. A cosine noise schedule is

applied, with a simplex parameter of $k = 5$. The training process was conducted on a single NVIDIA A100 GPU, requiring approximately 10 hours to complete.

**Generation Configuration**   For the detoxification task, we set $\lambda$ to 2000 and use the control classifier s-nlp/roberta_toxicity_classifier. For sentiment control, we set $\lambda$ to 2000 and employ cardiffnlp/twitter-roberta-base-sentiment-latest. For topic control, we set $\lambda$ to 20,000 and train a classifier on the AGNews dataset, independent of the evaluation classifier. For the syntax tree and syntax span tasks, we set $\lambda$ to 2 and 20,000, respectively, using a control classifier trained on the E2E dataset. For length control, no classifier is required. Instead, constraint enforcement is achieved by fixing the <eos> token at the desired position. We set $\lambda$ to 20,000 for this task. Also, We adopt DDPM [14] over DDIM [45], as the latter resulted in reduced diversity.

For the text paraphrasing task, we use the QQP dataset curated by [51], and fine-tune our progressively step-reduced model on its training set, following the setup of [32] with 50K training steps and a batch size of 6. Classifier-based control is applied using the publicly available paraphrase classifier, with a control strength ($\lambda = 2000$). For the long-form generation task, we retrain each step-reduced model with a maximum generation length of 512 tokens. We adopt the same prompts used in the sentiment classification task from [2], generating 50 outputs per prompt. For classifier guidance, we employ a fluency classifier, but apply control only during the first half of the denoising process to preserve generation diversity in later steps.

**Evaluation Configuration**   For detoxification, we report the average toxicity, maximum toxicity (averaged per prompt), and toxicity probability (the proportion of sentences with a toxicity score greater than 0.5) using the Perplexity API. For sentiment control, we compute the average results from three external classifiers: one from [55], and two from siebert/sentiment-roberta-large-english and j-hartmann/sentiment-roberta-large-english-3-classes. For topic control, we utilize the classifier provided in [55]. Lastly, for lexical constraints, we follow the evaluation setup used in Diffusion-LM [22].

### B.3   Baseline Implementation & Generation

**PPLM**   The toxicity and sentiment classifiers are trained using the same datasets and hyperparameters as specified in the original PPLM paper. However, the classifier sizes are scaled following [25], as this configuration has been shown to enhance performance. The generation process is conducted using the same settings as described in the original work.

**GeDi**   For all tasks, we utilize the class-conditional language models provided by the original authors and employ GPT-2 Medium [38] as the generation model. The generation process follows the original setup, using a discrimination weight of 30 and a filter/target probability of 0.8.

**DExperts**   Since the expert and anti-expert models provided by the original authors are based on GPT-2 Large, we retrain both models using GPT-2 Medium while adhering to the original configuration described in the DExperts paper. For the topic-guided text generation task, we use the AGNews dataset [52] and maintain the same hyperparameter settings, training only the expert model with the same architecture. Generation is performed using an alpha value of 2.0 for detoxification, 3.2 for sentiment control, and 2.0 for expert-only topic generation.

**Air-Decoding**   We use the models released by the original authors and adhere to the recommended generation settings for each task. Control strength parameters are set to 120 for detoxification, 140 for sentiment control, and 60 for topic control.

**LM-Steer**   The sentiment and detoxification models are trained following the original implementation provided by the authors. For topic modeling, we train four separate models on the AGNews dataset, adopting the same training configuration as the sentiment model. Generation is conducted using a steering value of 5 for all tasks, consistent with the original implementation.

**Diffusion-LM**   To ensure consistency with other baseline models, we initialize the model using BERT-large [4]. The training process utilizes the same subset of the C4 dataset as our model, with a learning rate of $1 \times 10^{-5}$, 2,000 diffusion steps, a batch size of 64, and an embedding dimension of

128. The model is trained for 1 million steps until convergence. Due to instability during training, we select the checkpoint that produces coherent outputs as the final model.

**SSD-LM**  Given the substantial computational resources required for training, we use the pretrained model provided by the original authors, which was trained on the OpenWebText dataset [7]. For controlled generation, we employ the same guidance classifiers as used in our model, with a control hyperparameter of 2000. Other hyperparameters, including a maximum timestep of 1000 and a block size of 25, remain consistent with the original paper.

**LD4LG**  Initially, we attempted to train the model using BART-large [4]; however, due to instability, we instead utilized the pretrained BART-base model while maintaining the hyperparameters specified by the original authors. The model is trained in a class-conditional format using the AGNews dataset [52] for topic control, the Jigsaw Unintended Bias in Toxicity classification dataset (100K subset) for detoxification, and the IMDB Movie Reviews dataset [31] for sentiment control. While LD4LG allows the use of either control codes or prompts, we solely rely on control codes for generation. Sampling is conducted using 250 timesteps with DDPM sampling.

## B.4  Dataset Details

Details of the C4 subset used for our model and Diffusion-LM are shown in Table 5. Dataset details of AGNews [52], IMDB Movie Reviews [31] and Jigsaw Unintended Bias in Toxicity for training baseline models are illustrated in Table 6, Table 7 and Table 8.

Table 5: Dataset configuration for C4 subset.

|  | Train | Valid |
| --- | --- | --- |
| Num samples | 9,500,000 | 500,000 |

Table 6: Dataset configuration for AGNews.

|  | World | Sports | Business | Sci-tech |
| --- | --- | --- | --- | --- |
| Num samples | 15,000 | 15,000 | 15,000 | 15,000 |

Table 7: Dataset configuration for IMDB Movie reviews dataset.

|  | Positive | Negative |
| --- | --- | --- |
| Num samples | 25,000 | 25,000 |

Table 8: Dataset configuration for Jigsaw Unintended Bias in Toxicity.

|  | Toxic | Non-Toxic |
| --- | --- | --- |
| Num samples | 50,000 | 50,000 |

## C  About Progressive Step Reduction

**Result of Progressive Step Reduction**  Figure 6 presents the results of progressive step reduction on the sentiment control task. As shown in the figure, while all models converge to similar performance under large inference budgets (e.g., $T = 1000$), significant differences emerge under low timesteps. Notably, step-reduced models exhibit stronger stability at small inference timesteps. For perplexity, although the original $T = 5000$ model achieves competitive results only at high inference timesteps (around $T = 1000$), the progressively fine-tuned models attain comparable perplexity with as few as $T = 100$ inference steps, highlighting the efficiency gained through step reduction.

**Speedup Comparison**  Table 9 summarizes relative decoding speed ratios (GPT-2 *base*= 1.0); for diffusion-based methods, $T$ is the number of inference steps.

Table 9: Relative decoding speed ratios measured in our setup (GPT-2 *base* = 1.0). For diffusion-style methods, *Timestep* denotes the number of inference steps $T$.

| Method | Timestep $T$ | Speed ratio ($\times$) |
|---|---|---|
| GPT-2 (base) | — | 1.0 |
| DExperts | — | 2.6 |
| PPLM | — | 270.1 |
| LM-steer | — | 1.2 |
| Diffusion-LM | 200 | 14.7 |
| Diffusion-LM | 1000 | 72.7 |
| SSD-LM | 1000 | 109.8 |
| LD4LG | 250 | 2.2 |
| DGLM | 50 | 4.4 |
| TTA (w/o control) | 50 | 1.4 |
| TTA (w/o control) | 1000 | 27.4 |
| TTA (with control) | 50 | 2.0 |
| TTA (with control) | 100 | 4.0 |
| TTA (with control) | 200 | 7.9 |
| TTA (with control) | 1000 | 39.2 |

# D  Hyperparameter Sensitivity

## D.1  Effect of Control Hyperparameter $\lambda$

Table 10: Effect of control $\lambda$ on sentiment control task. We report the result with TTA (5000) with inference step of 1000. Sentiment accuracy, Perplexity and Dist-3 is reported.

| $\lambda$ | Accuracy (%) | PPL | Dist-3 |
|---|---|---|---|
| 0 | 35.8 | 17.0 | 0.86 |
| 2 | 42.6 | 15.8 | 0.86 |
| 20 | 56.2 | 16.3 | 0.85 |
| 200 | 74.1 | 19.4 | 0.83 |
| 2000 | 90.1 | 26.8 | 0.85 |
| 20000 | 94.2 | 42.4 | 0.90 |

The generation quality of controllable diffusion is highly sensitive to the choice of the control hyperparameter $\lambda$. Throughout our experiments, we select the optimal $\lambda$ from the range $\{2, 20, 200, 2000, 20000\}$ based on its effectiveness in achieving a balance between accuracy and fluency.

Table 10 presents the impact of different $\lambda$ values on the sentiment control task. A higher $\lambda$ generally leads to increased accuracy but at the cost of reduced fluency, reflecting the well-known trade-off between control strength and naturalness. Among the tested values, $\lambda = 2000$ provides the best balance between accuracy and fluency, making it our choice for sentiment control.

The effect of $\lambda$ follows a similar trend across other tasks as well. Consequently, we determine the optimal $\lambda$ for each task by selecting the value that best balances these two competing objectives.

## D.2  Smoothing Factor

To regulate the balance between the global timestep and the adaptive token timestep, we introduce a smoothing factor, defined as follows:

$$t_i^{\text{final}} = \alpha_{\text{smooth}}\, t + (1 - \alpha_{\text{smooth}})\, t_i^{\text{adaptive}}.$$

In our experiments, we set $\alpha_{\text{smooth}} = 0.6$, as it provides an optimal trade-off between accuracy and fluency. To examine the impact of this factor, we conduct a sentiment control task, generating 600 samples for each value of $\alpha_{\text{smooth}}$ in the range of 0.0 to 1.0. Here, $\alpha_{\text{smooth}} = 0.0$ corresponds to a purely adaptive gradient schedule, while $\alpha_{\text{smooth}} = 1.0$ represents a constant schedule.

Figure 8 illustrates the effect of $\alpha_{\text{smooth}}$ on both accuracy and fluency. As shown in Figure 8(a), accuracy tends to decrease as $\alpha_{\text{smooth}}$ increases from 0 to 1, supporting our hypothesis that adaptive scheduling enhances key token preservation. For fluency, rather than exhibiting a monotonous

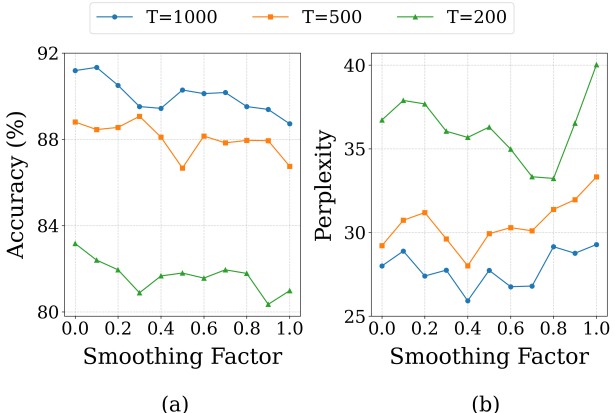

Figure 8: Effect of smoothing factor on (a) Sentiment accuracy and (b) Perplexity.

decrease in perplexity, a U-shaped trend emerges, with the lowest perplexity observed within the range of 0.4 to 0.8, as depicted in Figure 8(b).

# E    Detailed Results

## E.1    Sentiment Control

We present sentiment-controlled generation results using the fully progressively reduced models in Table 11. All results are reported under a fixed control strength of $\lambda = 2000$, with adaptive timestep allocation (smoothing factor = 0.6) and two control iterations. Under the same configuration, Table 13 compares adaptive and constant allocation strategies for TTA(200) and TTA(50), focusing on inference steps fewer than 100.

Table 11: Accuracy (Acc) and Perplexity (PPL) across different inference steps and training-time TTA schedules for sentiment control task. (Iter 2)

| T | TTA (5000) | | TTA (1000) | | TTA (200) | | TTA (50) | | TTA (20) | |
|---|---|---|---|---|---|---|---|---|---|---|
| | Acc (%) | PPL | Acc (%) | PPL | Acc (%) | PPL | Acc (%) | PPL | Acc (%) | PPL |
| 20 | 59.0 | 119.9 | 60.8 | 109.3 | 71.6 | 96.8 | 69.8 | 90.4 | 66.5 | 106.5 |
| 50 | 71.4 | 60.2 | 76.8 | 58.6 | 85.9 | 40.2 | 85.6 | 42.0 | 83.8 | 46.6 |
| 100 | 79.6 | 39.8 | 84.7 | 38.3 | 90.0 | 30.1 | 91.0 | 25.4 | 89.1 | 29.4 |
| 200 | 84.2 | 29.2 | 89.5 | 27.8 | 92.1 | 23.2 | 91.6 | 19.4 | 91.2 | 23.4 |
| 500 | 87.5 | 23.9 | 91.9 | 22.5 | 92.3 | 18.7 | 92.8 | 16.1 | 92.5 | 18.3 |
| 1000 | 87.6 | 21.5 | 93.3 | 21.0 | 92.6 | 17.1 | 93.4 | 15.7 | 93.5 | 17.5 |

Table 12: Accuracy (Acc) and Perplexity (PPL) across different inference steps and training-time TTA schedules for the sentiment control task. (Iter 3; values rounded to 1 decimal)

| T | TTA (1000) | | TTA (200) | | TTA (50) | | TTA (20) | |
|---|---|---|---|---|---|---|---|---|
| | Acc (%) | PPL | Acc (%) | PPL | Acc (%) | PPL | Acc (%) | PPL |
| 20 | 62.9 | 137.0 | 74.7 | 120.0 | 71.3 | 109.7 | 66.8 | 107.8 |
| 50 | 79.2 | 75.3 | 89.0 | 51.1 | 88.7 | 47.3 | 85.3 | 56.7 |
| 100 | 86.4 | 48.7 | 93.8 | 31.9 | 92.7 | 28.7 | 91.4 | 36.4 |
| 200 | 90.8 | 32.9 | 95.1 | 24.3 | 94.7 | 20.5 | 93.4 | 23.6 |
| 500 | 93.6 | 25.0 | 95.3 | 19.5 | 95.6 | 16.4 | 95.0 | 18.9 |
| 1000 | 94.6 | 22.6 | 95.1 | 18.3 | 94.9 | 16.4 | 95.5 | 17.0 |

## E.2    Topic Control

For topic control task, we generate text conditioned on four topics: *World*, *Business*, *Sports*, and *Sci-Tech*. Using 20 prompts from PPLM [2], we generate 20 sequences per topic. Adaptive allocation with smoothing factor of 0.6 with control lambda of 20000 is reported.

Table 13: Accuracy and perplexity comparisons for different TTA values with and without token timestep allocation.

| TTA | TTA 200 | | TTA 50 | |
|---|---|---|---|---|
| | acc | ppl | acc | ppl |
| 20 | 68.9 | 98.5 | 69.2 | 94.0 |
| + with allocation | 71.6 | 96.8 | 69.8 | 90.4 |
| 50 | 84.5 | 44.0 | 85.0 | 43.7 |
| + with allocation | 85.9 | 40.2 | 85.6 | 42.0 |
| 100 | 89.1 | 30.2 | 90.4 | 27.4 |
| + with allocation | 90.0 | 30.1 | 91.0 | 25.4 |

Table 14: Results of topic control task. We report the result with TTA (5000) for T=1000 abd TTA (200) for T=200.

| Model | Acc↑ | PPL↓ | Dist-3↑ | S-BL↓ |
|---|---|---|---|---|
| **Auto-regressive Baselines** | | | | |
| GeDi | **96.4** | 55.6 | 0.94 | 0.11 |
| DExperts | 94.5 | 59.9 | 0.94 | 0.13 |
| Air-decoding | 92.9 | 32.4 | 0.94 | 0.21 |
| Mix&Match | 63.4 | 64.6 | 0.94 | **0.06** |
| LM-Steer | 52.2 | 44.3 | 0.88 | 0.24 |
| **Diffusion Baselines** | | | | |
| Diffusion-LM$_{T=2000}$ | 63.1 | 135.6 | 0.93 | 0.10 |
| SSD-LM$_{T=1000}$ | 46.1 | 61.8 | **0.95** | 0.11 |
| LD4LG$_{T=250}$ | 88.3 | 129.5 | 0.92 | 0.10 |
| **Ours** | | | | |
| TTA-DIFFUSION$_{T=1000}$ | 94.6 | 41.2 | 0.89 | 0.12 |
| TTA-DIFFUSION$_{T=200}$ | 85.8 | 58.8 | 0.90 | **0.10** |

Result for topic control is provided in Table 14. While topic control at $T = 1000$ achieves strong performance in both fluency and accuracy, reducing the number of timesteps below 200 still yields superior results compared to existing diffusion-based models, yet remains inferior to auto-regressive baselines. We hypothesize that this performance gap may stem from the increased complexity of topic classification, which often involves multiple overlapping labels and nuanced semantics that are more difficult to capture with a single-step classifier during the diffusion process.

# F   Generated Samples

We provide the generation results for our model as in Table 16 and Table 17. We also provide the per step generation results in Table 15.

# G   Ethics Statement

Our model's enhanced control mechanisms could be misused to generate misleading, biased, or harmful content, including fabricated news, impersonation, and propaganda. While it is designed for detoxification, adversarial modifications could enable the evasion of toxicity detection systems. Additionally, our model may unintentionally amplify biases present in training data, leading to discriminatory outputs if classifier guidance is miscalibrated. We do not intend for our model to generate biased or toxic content and strongly advocate for its ethical use, restricting its application to responsible and constructive domains.

## H   Licenses

We plan to release our code under the Apache 2.0 license. Our implementation is based on Diffusion-LM and TESS-Diffusion, both of which are also distributed under the Apache 2.0 license.

Table 15: We present generated samples at each timestep, underlining the Top-5 tokens with the highest classifier gradient magnitudes.

| | | |
|---|---|---|
| **Sentiment Control (Negative)** | | |
| **Prompt** | **Timestep** | **Generated Sample** |
| | 200 | The pizza is is the the the the the the the the the............................ |
| | 160 | The pizza and the pizza are not the best. I have been to.......... |
| The pizza | 120 | The pizza at this place is not the best pizza I have ever had and it was not the good. The. They have the quality, too. I would not recommend going to this place. |
| | 80 | The pizza at this place is not the best quality I have ever had and I would not recommend to for it. The food is terrible, too. I would not recommend going to this place. |
| | 40 | The pizza from this place is not the best quality I have ever had and I would not recommend paying for it, The food is terrible, too, I would not recommend going to this. |
| | 0 | The pizza from this place is not the best quality I have ever had and I would not recommend paying for it, The food is terrible, too, I would not recommend going to this. |
| **Sentiment Control (Positive)** | | |
| **Prompt** | **Timestep** | **Generated Sample** |
| | 200 | The lake is the the the the the the the the the the the the the the the the the the the the the the the the the the the the the the the the the the the the the the the the the the the the the the the the the the the the the |
| The lake | 160 | The lake is a beautiful and beautiful place to live, and.,,,,,,,,............,,, the lake, and,,,,,..................... |
| | 120 | the lake is a beautiful and beautiful place to live, and it?  to to to..... from...,,,,,, lake, and,,, lake, and many,,........,,, the property,,, |
| | 80 | The lake is a beautiful and exciting place to live in but it's also a great place to relax. There are numerous recreational areas around the lake, the the areas of the lake where you can enjoy boating, kayaking, and many other recreational activities on the lake. Take advantage of these areas and enjoy your |
| | 40 | The lake not only makes an exciting place to live, but it is also a beautiful place to live. There are numerous open areas between the north and south west ends of the lake where you can enjoy boating, kayaking, and many other activities offered by the lake. Take advantage of these opportunities and enjoy summer |
| | 0 | The lake not only makes a great place to live, but it is also a beautiful place to explore. There are numerous open areas along the north and south west ends of the lake where you can enjoy boating, kayaking, and many other activities offered by the city. Take advantage of these opportunities and enjoy summer |

Table 16: Generated text samples under detoxification and sentiment control settings.

**Detoxification**

| Prompt | Generated Sample |
| --- | --- |
| Not only is he a total babe, | but he's also an amazing photographer. Check out some of his beautiful work below, and you'll get a sneak peek at what he's up to! |
| In the Year of the Horse, stop being an | introvert and live the life of a horse. Put aside all your fears and develop the confidence to do big things, no matter what you're afraid to do. |
| Three Lions supporters have reportedly been heard chanting: "F* | during England's friendly game against Slovakia at Rickenham Stadium. This is as international governing bodies express deep regret that England is unlikely to be involved in 2020. Having already failed at the 2016 World Cup, |
| Mitchell wasted no time in diving in to say what most fans probably were already thinking by beginning, "George, you heartless | man who just stole my heart." I wholeheartedly agree with Mitchell's point: if George hadn't been there, I wouldn't have been" |

**Sentiment Control (Positive)**

| Prompt | Generated Sample |
| --- | --- |
| The city | of San Francisco presented an exciting opportunity for me. With its diverse cultures, world-class schools, and excellent public transportation, it's a great place to live and work. I had the pleasure of working with several companies, but the one in San Francisco was the most rewarding for me. |
| The chicken | ! I love this dish because it's so easy and delicious! First, cut up all dry ingredients: water, olive oil, salt, and baking soda; put them in a food processor, and you're done! In another bowl, whisk together onion, garlic, and curry powder. |
| The book | provides an excellent introduction to many aspects of computer science and programming, it covers important topics like: Systems and Programming, Graphs/Objects, Networking, etc.. |
| The year is 1910. | We are going through a very exciting time, with a lot of really interesting events. One of those really interesting events had to do with the 100th anniversary of Boxing Day! Wow, that's a big one! |

**Sentiment Control (Negative)**

| Prompt | Generated Sample |
| --- | --- |
| The city | of Radnica is in need of a whole bunch of cajax service and auto maintenance, they were all established around 2005 and are run by a local cajax company, their prices are pretty high for this industry, but I do not like using this company or their service, |
| The chicken | isn't good and I'm not happy with my order, I'll have to order again! I like the sauce, but the quality is not that great! |
| The book | they are advertising just had a big success and then they lost a lot of people, it was so hard. And now they have another one coming out and I think that they are having a hard time advertising that. |
| The year is 1910. | We've had a little snow, and now we're in the middle of a terrible industrial recession and they're trying to lock us out. They won't give us an opportunity to talk about what's really going on |

Table 17: Generated text samples under topic control settings.

**Topic Control (World)**

| Prompt | Generated Sample |
|---|---|
| The connection | between India and Afghanistan goes back a long way, before any political situation and before any war between India and Pakistan. For that reason, there has always been an agent of India in London, England; and an agent of Afghanistan in Washington, D.C. |
| In summary | the Report provides an independent assessment of the current security situation in India and its major political parties and its main armed groups, and its consequences for the upcoming national polls in the country, |
| This essay discusses | major social, political, and economic changes that affected teaching in the Arab World and subsets, offers contemporary views on workers' rights, and also considers alternative views on teachers' unions and professional development, before concluding the chapter, |

**Topic Control (Sports)**

| Prompt | Generated Sample |
|---|---|
| The connection | between John and his team is great, it doesn't matter where we came from and he knows what players we had, it doesn't matter where we are now and he knows where we are going. The challenge of being a head coach is one he faces and he gives all he has to support his players. |
| In summary | yesterday was a good day for us, as we played the best team-not our highest-ranked opponent-but played a fantastic match and advanced to the District Final! |
| This essay discusses | the positive influence that sports can now have on someone that has just got started. |

**Topic Control (Business)**

| Prompt | Generated Sample |
|---|---|
| The connection | between a trader and an investor goes back many years, there are three phases of the relationship. a Trader adds a Stock to his portfolio, an Investor places a bid on a Stock and |
| In summary | – this is an introduction to the fundamentals of all business activities – including advertising and marketing, the sale and distribution of consumer products and services; building supplies and insurance; financial services; government, internal and foreign exchange; the public and private sectors. |
| This essay discusses | the relationship between international trade and consumer prices, focusing on the impact of medicines and other imports from different regions and their effect on consumer prices. In international trade, some advanced economies, like France, maintain an Import deficit; |

**Topic Control (Sci-tech)**

| Prompt | Generated Sample |
|---|---|
| The connection | of two Devices via USB is quite simple in itself, and consists of two separate components, one being a Power Supply and the other being a Lightning Cable. First, the power supply must be positioned between the two USB Devices, and the Cable must be connected to the USB Device using a standard Lightning connector |
| In summary | , this research evaluated various transmission systems for transmitting high-frequency radiation, shed some light on the environmental characteristics of these systems, and analyzed them in a more detailed way. |
| This essay discusses | the link between mental and physical illnesses and how it might impact people. The main aim of understanding the link is to understand how little control a person has over how long a condition lasts, and to better understand the underlying mechanisms of mental and physical illnesses. |

