# OpenReview forum: "Don’t Let It Fade: Preserving Edits in Diffusion Language Models via Token Timestep Allocation"
_NeurIPS.cc/2025/Conference — NeurIPS 2025 poster_

### Official Review · Reviewer_NdQ8 · 2025-07-01

**Clarity:** 3
**Significance:** 2
**Originality:** 2
**Rating:** 4
**Confidence:** 3

**Summary:**

The paper investigates why classifier-guided diffusion language models become unstable, tracing the problem to two factors—excessive fluctuations and update-forgetting—and proposes adaptive per-token timestep allocation as an inference-time fix. Additionally, they bridge the discrete–continuous gap through logit-simplex mapping and uses timestep distillation to reduce sampling steps. Extensive experiments and ablations confirm that the method yields more stable and controllable generation.

**Questions:**

Please refer to the weakness section.

**Ethical Concerns:**

["NO or VERY MINOR ethics concerns only"]

**Final Justification:**

The main concern of my previous rating was the limited evaluation and the limited novelty of the proposed method but the author reloved most of it. Therefore I raise my rating.

**Limitations:**

Please refer to the weakness section.

**Quality:**

2

**Strengths And Weaknesses:**

Strengths
1. The paper’s clear empirical diagnosis (fluctuation-driven update-forgetting) naturally motivates the adaptive per-token timestep strategy, making the solution intuitive and clear.
2. The method shows consistent improvements in overall metrics compared to other baselines even with reduced sampling steps.
3. The author provides extensive experiments in the main paper and the appendix.

Weaknesses
1. While adaptive token-level timesteps for classifier-guided inference are a thoughtful extension, I feel this idea is incremental, as it builds on earlier token scheduling concepts like those in AR-Diffusion. Moreover, other components such as the logit-simplex mapping and timestep reduction have already been proposed in prior work.
2. From my understanding, the time distillation is used for compensating the time overhead from using the classifier guidance. I think there should be a comparison regarding the time consumption for generating the text.
3. Several informative studies in the appendix are not mentioned in the main paper making it easy for readers to miss evidence that actually strengthens the claims.

---

> ### Author Rebuttal · Authors · 2025-07-31
>
> **Deer reviewer NdQ8,**
>
> Thank you for your careful and constructive review. We value your thoughtful feedback and respond to your concerns below.
>
> ## **W1. Novelty of TTA-DIFFUSION**
>
> We appreciate your reflection on the novelty of our method. While we agree that certain components—such as logit-to-simplex mapping and timestep reduction—draw from existing practices and are primarily design choices for integrating classifier guidance and improving efficiency, we would like to emphasize that our core contribution lies in the **adaptive token-level timestep allocation** mechanism.
>
> This mechanism introduces a **principled, inference-time strategy** that dynamically adjusts the generation schedule of each token based on its alignment with an external semantic signal—measured via classifier gradients. In contrast to AR-Diffusion, which relies on fixed schedules tied to token position or heuristic progression, our method **actively infers** token importance at each step and tailors refinement accordingly. **This dynamic and semantics-aware scheduling is crucial for preserving intended control signals, reducing unnecessary updates, and enabling more stable, accurate, and efficient generation under guidance.**
>
> Importantly, our method directly addresses a core weakness in diffusion-based controllable generation: the phenomenon of *update-forgetting*, where classifier-guided semantic edits are inadvertently erased in later steps. By assigning smaller timesteps to classifier-aligned tokens, our approach effectively protects critical updates from being overwritten—**preserving semantic consistency over time**, a capability not supported by prior scheduling methods. Moreover, by focusing denoising effort on uncertain tokens, our method **prioritizes computation where it's most needed**, leading to significantly improved fluency and control even under aggressive step reduction.
>
> In addition, the plug-and-play nature of our method makes it broadly applicable across both discrete and continuous diffusion frameworks without requiring model retraining, making it a **scalable and general-purpose solution** for guided generation. To our knowledge, this is the first work to propose such an **adaptive, gradient-informed inference-time allocation strategy** in diffusion language modeling.
>
> We believe this formulation offers more than a technical distinction from AR-Diffusion—it introduces a scalable and theoretically grounded mechanism for **coordinating the generation order with external goals**, which is essential for tasks requiring reliable semantic control (e.g., detoxification, sentiment modulation, alignment). In doing so, our work both addresses core weaknesses in existing diffusion generation (e.g., update instability) and paves the way for future advances in guided generation under diffusion frameworks.
>
> ## **W2. Empirical Speedup**
>
> Thank you for the helpful suggestion. As you correctly pointed out, we introduced **time distillation** to mitigate the **latency overhead** caused by classifier guidance.
>
> To support this claim, we conducted a **latency comparison** across various methods using a unified evaluation setup: sequence length of 64, batch size of 100, evaluated on a single A100 GPU. Latency results for DExperts, PPLM, LM-Steer, and DGLM were drawn from [1] and [2].
>
> As shown in the table below, our TTA-Diffusion method (with classifier control) offers **significant runtime improvements** over existing classifier-guided diffusion approaches such as SSD-LM and Diffusion-LM, while remaining competitive with autoregressive baselines:
>
> **Empirical speed comparison**
>
> | Method | Timestep | Speed ratio |
> | --- | --- | --- |
> | GPT-2 (base) | — | 1.0 |
> | DExperts | — | 2.6 |
> | PPLM | — | 270.1 |
> | LM-steer | — | 1.24 |
> | Diffusion-LM | 200 | 14.7 |
> | Diffusion-LM | 1000 | 72.7 |
> | SSD-LM | 1000 | 109.8 |
> | LD4LG | 250 | 2.2 |
> | DGLM | 50 | 4.4 |
> | TTA (w/o control) | 50 | 1.4 |
> | TTA (w/o control) | 1000 | 27.4 |
> | **TTA (with control)** | 50 | 2.0 |
> | TTA (with control) | 100 | 4.0 |
> | TTA (with control) | 200 | 7.9 |
> | TTA (with control) | 1000 | 39.2 |
>
> This demonstrates that TTA-Diffusion effectively alleviates the computational burden of classifier guidance while preserving its controllability benefits. We have included this speed comparison in **Appendix C** to provide a clear and concrete evaluation of the latency benefits introduced by our method.
>
> [1] Han et al., “Word Embeddings Are Steers for Language Models”, ACL 2024
>
> [2] Lovelace et al., “Diffusion Guided Language Modeling”, Findings of ACL 2024
>
> ## **W3. Integration of Supporting Evidence**
>
> Thank you for highlighting this point. We agree that several **informative analyses** in the appendix warrant more explicit mention in the main paper.
>
> In response, we have updated the manuscript—particularly Sections 2 and 3.2—to better reference key supporting experiments, including:
>
> - The correlation between token fluctuations and evaluation metrics (Appendix B.1).
> - The rationale for adaptive timestep decisions (Appendix C).
> - Additional qualitative examples of update-forgetting and recovery (Appendix Table 12).
>
> Thank you for your constructive comments. We hope the updated results respond adequately to your concerns.

---

> > ### Author Response · Authors · 2025-08-05
> >
> > Dear Reviewer NdQ8,
> >
> > Thank you once again for your careful review and constructive observations. We appreciated your feedback, particularly around how our contributions are positioned and how supporting experiments are presented.
> >
> > As the author-reviewer discussion window comes to a close, we’d be happy to clarify any remaining points or questions you might have.
> >
> > Your suggestions were instrumental in helping us strengthen the clarity and presentation of our work.
> >
> > Warm regards,
> >
> > The Authors

---

> ### Comment · Reviewer_NdQ8 · 2025-08-05
>
> Thank you for your detailed answers; they have addressed my previous concerns. However, I do have one additional question.
>
> Since the core contribution of your work is the adaptive token timestep allocation, it would be important to directly compare the proposed adaptive schedule with the linear schedule mentioned in the main paper. I noticed that some accuracy comparisons on sentiment control are included in the appendix, but could you also provide a more comprehensive comparison across additional metrics and detoxification dataset?

---

> > ### Author Response · Authors · 2025-08-06
> >
> > Thank you for your follow-up and for highlighting the importance of a comprehensive comparison with the linear schedule baseline.
> >
> > We agree that including detailed results beyond the sentiment control accuracy reported in the appendix would provide a clearer picture of the effectiveness of our proposed adaptive method. To this end, we now include full metrics for both the **sentiment control** and **toxicity control** tasks, evaluated under varying timesteps (50, 100, 200). All results below are based on generation with 50 sampling steps. These comparisons cover a range of scheduling strategies, including our **adaptive schedule**, the **linear schedule**, and the **backward-linear (blinear) schedule**, alongside the default (no scheduling) baseline.
> >
> > - **Sentiment Control Results**
> >
> > | Timestep | Schedule | PPL ↓ | Accuracy ↑ | Dist-3 ↑ | Rept-3 ↓ |
> > | --- | --- | --- | --- | --- | --- |
> > | 50 | default | 44.0 | 83.5 | 0.87 | 6.8 |
> > |  | adaptive | **42.0** | **85.6** | **0.88** | 7.8 |
> > |  | linear | 49.8 | 82.1 | 0.88 | **6.7** |
> > |  | blinear | 48.0 | 74.8 | 0.85 | 9.6 |
> > | 100 | default | 27.8 | 89.4 | 0.86 | 12.1 |
> > |  | adaptive | **25.4** | **91.0** | **0.86** | 12.1 |
> > |  | linear | 29.1 | 89.7 | 0.86 | 12.3 |
> > |  | blinear | 31.2 | 83.1 | 0.83 | **12.0** |
> > | 200 | default | 22.9 | 92.8 | 0.84 | **12.1** |
> > |  | adaptive | **20.5** | **94.7** | **0.85** | 12.2 |
> > |  | linear | 24.5 | 89.3 | 0.84 | 12.3 |
> > |  | blinear | 23.7 | 87.3 | 0.81 | 14.9 |
> >
> > - **Toxicity Control Results**
> >
> > | Timestep | Schedule | PPL ↓ | Avg. Toxicity ↓ | Max. Toxicity ↓ | Toxicity Prob ↓ | Dist-3 ↑ |
> > | --- | --- | --- | --- | --- | --- | --- |
> > | 50 | default | 68.0 | 14.0 | 29.0 | 2.21 | 0.94 |
> > |  | adaptive | 61.2 | 12.7 | **27.1** | **1.64** | 0.94 |
> > |  | linear | **59.5** | **12.5** | 27.3 | 1.95 | 0.93 |
> > |  | blinear | 127.6 | 14.3 | 30.3 | 2.43 | **0.96** |
> > | 100 | default | 48.2 | 13.8 | 28.7 | 2.22 | 0.93 |
> > |  | adaptive | 46.8 | 12.3 | 26.9 | 2.08 | 0.93 |
> > |  | linear | **46.3** | **12.2** | **26.7** | **1.78** | 0.93 |
> > |  | blinear | 78.4 | 13.6 | 29.2 | 2.12 | **0.94** |
> > | 200 | default | 41.4 | 13.6 | 28.6 | 2.17 | **0.93** |
> > |  | adaptive | **40.1** | **12.2** | **25.4** | **1.78** | 0.92 |
> > |  | linear | 40.6 | 12.2 | 26.0 | 1.96 | 0.92 |
> > |  | blinear | 65.1 | 13.1 | 27.8 | 2.21 | 0.93 |
> >
> > These results demonstrate that the adaptive schedule consistently outperforms other schedules in sentiment control tasks. For toxicity control, while the linear schedule performs comparably to the adaptive one, both significantly outperform the default (no schedule) baseline in terms of toxicity reduction and perplexity.
> >
> > We have included these detailed results in **Appendix C** to ensure clarity and reproducibility.
> >
> > We hope this comprehensive comparison addresses your request and further supports the contribution of our proposed scheduling strategy.

---

### Official Review · Reviewer_ThEE · 2025-07-03

**Clarity:** 3
**Significance:** 2
**Originality:** 3
**Rating:** 3
**Confidence:** 3

**Summary:**

* The authors propose TTA-Diffusion, an inference-time technique to reduce token fluctuations when applying classifier guidance.
* The authors observe that semantics of generated sequences are perturbed and modified during diffusion and formalize this problem as diffusion fluctuation and update-forgetting.
* The authors propose a dynamic timestep allocation strategy that leverages gradient signals from the guiding classifier to ensure that recently guided tokens are not immediately revered or perturbed by the denoising model.
* The authors show that TTA-Diffusion outperforms autoregressive and diffusion baselines on controllability tasks, such as detoxification, sentiment control, topic control, etc.

**Questions:**

Comments and questions are not necessarily in order of importance (largely chronological with the flow of the paper).

1. Are the examples in Figure 1 actual observed samples / examples of update-forgetting? The transition seems drastic. Diffusion disruption and update-forgetting are defined by strict token-equality checks, which seem brittle and susceptible to over-estimation due to minor word substitution to synonyms or modifications to sentence structure. Actual observed samples from the experiments in Appendix B.1 would be helpful in contextualizing the problem (maybe alongside TTA-Diffusion outputs in Table 12 for clearer comparison).

2. The authors write that TTA-Diffusion "build[s] upon the Simplex-based diffusion language model [11], and extend it into a fully non-autoregressive framework [28] that naturally supports external classifier guidance." However, the non-autoregressive extension simply follows from using TESS, so it seems slightly misleading to suggest this as a contribution. A rephrasing seems necessary.

3. It would be helpful if Definition 2 includes details on how key tokens from classifier guidance are identified (highest gradient norm, as written in Hypothesis 2. Moreover, Table 12 seems to suggest that context / prompt tokens are included in computing the top $K$ most important tokens -- is this correct? I would expect context tokens to be excluded from this measurement as well as updates from classifier guidance or diffusion.

4. Sec. 2.2. indicates that TESS without self-conditioning seems to be used. Does enabling self-conditioning make a difference? The simplex averaging self-conditioning method is effectively an exponential moving average of predicted probabilities, so I would expect the predictions to remain more consistent without excessive disruptions.

5.  At a high level, it seems like update-forgetting occurs because the backward diffusion step erases the effect of classifier guidance. Is the guidance factor at all relevant to this? Update-forgetting seems to suggest that classifier guidance was perhaps too aggressive that it causes a "tug-of-war" between guidance updates and backward diffusion. Can a well-tuned guidance factor alleviate this? It seems like $\alpha_\mathrm{smooth}$ explicitly modulates this, but really a fine-tuned $\lambda$ should suffice (and perhaps the adaptive timestep allocation strategy in Sec. 3.2 is effectively controlling $\lambda$ per timestep per token).

6. Hypothesis 1 seems to suggest token fluctuations is the cause of reduced fluency and coherence. Is it possible that the disruptions are due to model's low confidence, i.e., the reduced fluency is more the byproduct of the model struggling to converge on a stable output sequence due to the difficulty of generating a sample? From this point of view, the reduced fluency is not due to the fluctuations per se, but due to low confidence / more uncertainty in the model's predictions?

7. Is the adaptive timestep in Sec. 3.2. have a monotonicity property, i.e., $f(i, t) < f(i, t + 1)$? It seems unnatural and contradictory for a token that was assigned a small timestep ("this is a stable, finished token") to later have a larger timestep ("now we need to aggressively refine"). The norm-based convex combination with time doesn't seem to guarantee this.

8. How does progressive step reduction relate to the earlier hypotheses and problem definitions? It's not clear to me what the motivation for this is, especially given the narrative that TTA-Diffusion is an inference-time technique.

9. Is update-forgetting unique to the continuous-to-discrete nature of simplex modeling, or do you think there are continuous analogues?

**Ethical Concerns:**

["NO or VERY MINOR ethics concerns only"]

**Limitations:**

yes

**Quality:**

2

**Strengths And Weaknesses:**

Strengths:
* Diffusion generation, and in particular timestep allocation strategies and fine-grained steering, is a relevant topic of interest to the community.
* Presented claims and hypotheses are empirically appropriately supported by experiments and ablations.
* The paper is clean and easy to follow.

Weaknesses:
* The paper focuses on a rather specific problem (steering simplex diffusion models via classifier guidance), limiting its general applicability.
* The definition of token disruption and update-forgetting seems rather simple and is unintuitive. It is based on strict token matching, which at a glance seems like a poor measure of semantic coherence.
* Some claims are weakly supported by theoretical justifications drawn from the discrete diffusion literature. While the rough intuition may translate to simplex diffusion models, some details and rigor are likely lacking.

---

> ### Author Rebuttal · Authors · 2025-07-31
>
> **Dear Reviewer ThEE,**
>
> Thank you very much for your thoughtful and detailed feedback. We greatly appreciate the time and effort you dedicated to reviewing our work. We address each of your comments and suggestions in the responses below.
>
> ## **W1. Applicability of TTA-DIFFUSION in various domains**
>
> We appreciate your concern about generality. While our main experiments use simplex-based diffusion models for their compatibility with classifier gradients, our adaptive  allocation strategy is broadly applicable and extends to both discrete and continuous diffusion frameworks.
>
> To demonstrate this, we integrated our method into the Discrete Diffusion Guidance (D-CBG) framework [1]. Please refer to our response to Reviewer X8kS (Q2) for further details.
>
> The same strategy also applies to continuous diffusion models. For example, applying it to a sentiment control task with Diffusion-LM [2] improved accuracy from 72.8% to 75.6%, demonstrating the method’s versatility and effectiveness across both discrete and continuous frameworks.
>
> [1] Schiff et al., “Simple Guidance Mechanisms for Discrete Diffusion Models”, ICLR 2025.
>
> [2] Li et al., “Diffusion-LM Improves Controllable Text Generation”, NeurIPS 2022
>
> ## **W2. Definition of token disruption and update-forgetting**
>
> Thank you for raising this important point. We understand the concern that our definitions of **token disruption** may appear too literal for capturing true semantic drift. To make our analysis more resilient to paraphrases and word-order variations, we are adding *semantics-aware* alternatives that complement the strict proxy:
>
> - **BLEU (n-gram overlap):** Captures surface-level similarity, tolerating minor word reordering.
> - **BERTScore (embedding-based):** Detects semantic shifts, including synonyms and paraphrases.
> - **Soft token matching:** Measures token-level semantic similarity via embeddings.
>
> We observe a moderate correlation between BLEU, BERTScore, and cosine similarity with perplexity at time *t* (−0.3∼−0.6), with trends mirroring the perplexity curves in Figure 2. BLEU shows the highest correlation, likely due to heavy input noise making n-gram overlap more informative than sentence-level embeddings.
>
> Our token-level fluctuation ratio also shows strong correlation with these metrics (e.g., BLEU −0.894, BERTScore −0.906), supporting its use as a practical proxy for semantic drift. However, to better capture subtler shifts such as paraphrasing or synonym substitutions, we have enhanced our evaluation by integrating these semantics-aware metrics.
>
> ## **W3. Theoretical justifications**
>
> Thank you for the valuable feedback. We agree that a clear theoretical link to discrete diffusion is important. Below is a three-part proof sketch supporting our claims; full derivations will appear in the revised manuscript (Appendix A.2–A.5).
>
> **(1) Relation of Simplex Diffusion to Discrete Diffusion**
>
> Our simplex-based diffusion model is formally connected to discrete diffusion frameworks via **Diffusion Duality** [1], which maps Gaussian diffusion in continuous space to a Uniform-state Discrete Diffusion Model (USDM). The core duality lies in projecting the continuous latent $X_t$ to a discrete token. While the theory uses $\arg\max$, we apply softmax with top-p sampling, which converges to $\arg\max$ as the threshold nears zero—providing a controllable generalization. This links our simplex diffusion model to the discrete diffusion framework. We have revised Appendix A.5 for this.
>
> **(2) Relation Between Diffusion Fluctuation and Perplexity**
>
> As detailed in Appendix A.2, our claim that higher diffusion fluctuation leads to increased perplexity is supported by the cross-entropy bounds in Haxholli et al. [2]. Fluctuation $R_t$ approximates sequence-level Hamming distance, and when the learned kernel $p_\theta$ deviates from the true kernel $q_t$, it assigns more mass to sequences with larger token shifts, increasing total variation. By Pinsker’s inequality, this leads to higher $\mathrm{KL}(q_t | p_\theta)$, relaxing the lower bound on cross-entropy and perplexity. Full derivations are in Appendix A.2–A.3.
>
> **(3) Theoretical Grounding of Adaptive Allocation**
>
> Our adaptive allocation strategy achieves a near-Pareto improvement by enhancing control accuracy without increasing perplexity. This is supported by two insights: (1) under an exchangeable reverse diffusion kernel, the perplexity bound ($J_2$) depends only on the multiset of per-token variances ${\sigma_i^2(t)}$, not their assignments, making perplexity invariant to token-wise permutations (Appendix A.3, Lemma 1); and (2) since classifier control is sensitive to noise distribution, we set $\sigma_i^2 \propto (1 - \hat{g}_i)$ to allocate less noise to well-aligned tokens and more to misaligned ones. This targeted variance reshaping reduces drift where accuracy is critical and promotes updates where needed.
>
> [1] Sahoo et al.,”The Diffusion Duality”, ICML 2025.
>
> [2] Haxholi et al., “Efficient Perplexity Bound and Ratio Matching in Discrete Diffusion Language Models”, ICLR 2025
>
> ## **Q1. Empirical observation of update-forgetting phenomena in Figure 1**
>
> Thank you for this question.  Yes, the examples in Figure 1 (e.g., “clean → dirty,”) are actual observations extracted from our generation outputs. While extreme sentiment reversals are uncommon, we frequently observe more subtle degradations (guided tokens becoming neutralized) across denoising steps.
>
> We also included more concrete examples from Appendix B.1 in the revised version.
>
> ## **Q2. Clarification on the contribution of TESS integration**
>
> We agree that the phrasing may have overstated our contribution in this regard. The non-autoregressive property of our framework indeed follows directly from adopting the TESS configuration. We have revised Section 3.3 of the manuscript to clarify that our work builds on the TESS-based setup.
>
> ## **Q3. Details on identifying key tokens**
>
> Thank you for pointing this out. We have revised *Definition 2* to clarify that **key tokens** are selected based on their **gradient norm magnitude**.
>
> For the prompt tokens, while Hypotheses 1 and 2 are unaffected (as they use unconditional generation), you are correct that we included prompt tokens when computing important token indices during inference. In our case, prompts were short and had minimal impact, but we acknowledge that longer prompts could introduce bias.
>
> Thanks to your suggestion, we have updated our inference algorithm to exclude prompt tokens from the importance calculation, ensuring robustness across different prompt lengths.
>
> ## **Q4. Effect of self-conditioning**
>
> We appreciate the insightful question. In our experiments, enabling self-conditioning had minimal impact on control metrics. While self-conditioning acts as a smoothing mechanism , we conjecture that its effect diminishes when classifier guidance is used.
>
> ## **Q5. Relevance of update-forgetting to guidance factor**
>
> This is an excellent point. We agree that update-forgetting arises from the tension between classifier guidance and denoising, particularly when guidance is overly strong. As Table 8 shows, a high guidance scale raises perplexity, suggesting that excessive guidance can destabilize generation by overwriting updates. Our adaptive timestep allocation implicitly adjusts the local guidance scale $λ_i$ by assigning each token a timestep $t_i$ based on its gradient. Since smaller $t_i$ weakens denoising ($|G_d|$), this is equivalent to scaling the update from $G_d(t) + λ G_c$ to $s_i G_d(t) + λ G_c$, or $G_d(t) + λ_i G_c$ with $λ_i = λ / s_i$.
>
> ## **Q6. Cause of token fluctuations**
>
> Thank you for the insightful hypothesis. While we observe a clear link between token fluctuation and reduced fluency, we agree that low model confidence may also contribute to instability. Our analysis shows a moderate negative correlation ($r = -0.33$) between confidence and next-step fluctuation, suggesting low-confidence tokens are more likely to change—though not strongly enough to imply causation. We view confidence and fluctuation as related but distinct signals of instability, reflecting uncertainty, gradient misalignment, and overwriting during denoising.
>
> ## **Q7. Monotonicity of adaptive timestep scheduling**
>
> Thank you for raising this point. We accounted for monotonicity by optionally enforcing a constraint that prevents a token’s timestep from increasing, but found it had minimal impact—typically within ±1% in metrics. Interestingly, the unconstrained version often performed slightly better, likely because (1) norm-based allocation is nearly monotonic in practice, and (2) early-step randomness lets initially irrelevant tokens later align with the target attribute, making re-refinement beneficial.
>
> ## **Q8. Rationale of progressive step reduction**
>
> Thank you for the question. The progressive step reduction is introduced primarily for practical reasons, rather than as a core theoretical component of our method. Since classifier guidance adds considerable computational overhead during inference, we adopt progressive step reduction to compensate for this cost.
>
> ## **Q9. Update-forgetting in continuous vs. simplex diffusion**
>
> We appreciate your thoughtful question. While update-forgetting is harder to characterize in continuous diffusion models, we bet it is more pronounced in simplex or discrete settings due to the stepwise projection toward token modes, which can override previous guidance. In contrast, continuous models evolve more smoothly without such hard projections, potentially reducing forgetting. Nonetheless, we observe similar forgetting dynamics across both settings, as applying our adaptive allocation to Diffusion-LM still yielded improved control.
>
>
> **We hope these new findings resolve your concerns. Thank you again for your helpful feedback.**

---

> > ### Author Response · Authors · 2025-08-05
> >
> > Dear Reviewer ThEE,
> >
> > We truly appreciate the time and thoughtful analysis you brought to your review. Your insights were especially helpful in guiding us to clarify theoretical aspects and refine our framing.
> >
> > As the discussion period winds down, we wanted to follow up in case there are any remaining concerns or questions you’d like us to address. We’re happy to provide further clarification if needed.
> >
> > Thank you again for your engagement and valuable feedback.
> >
> > Warm regards,
> >
> > The Authors

---

### Official Review · Reviewer_X8kS · 2025-07-03

**Clarity:** 3
**Significance:** 2
**Originality:** 3
**Rating:** 4
**Confidence:** 3

**Summary:**

The authors propose to improve the stability of classifier-guided semantic edits in diffusion language models by proposing context-dependent token updates. They show that their approach significantly improves both computational efficiency and generation quality.

**Questions:**

- Can the authors report the empirical speedups of proposed approach under fewer sampling steps?
- Can the context-dependent token updates be applied to existing discrete diffusion frameworks? [1]

[1] Schiff, Y. "Simple Guidance Mechanisms for Discrete Diffusion Models." ICLR 2025.

**Ethical Concerns:**

["NO or VERY MINOR ethics concerns only"]

**Final Justification:**

After the rebuttal discussion, I keep my score of "4".

- The proposed framework is novel with strong empirical results
- The authors addressed my concerns in missing baselines for discrete diffusion guidance and showing empirical speedups
- However, their approach does not achieve a speedup relative to AR under lower sampling steps as mentioned in their abstract.

**Limitations:**

yes

**Paper Formatting Concerns:**

no formatting concerns identified

**Quality:**

3

**Strengths And Weaknesses:**

**Strengths**
- The authors clearly formulate the "update-forgetting" failure mode with supporting empirical evidence and thorough ablations
- The proposed context-dependent token updates leads to significant improvements in both computational efficiency and generation quality
- The range of tasks and ablations are thorough and convincing

**Weaknesses**
- Empirical speedups of proposed approach under fewer sampling steps are not presented
- Proposed approach is grounded in Gaussian diffusion. However, SOTA diffusion LMs operate in a discrete state space, with recent work proposing guidance mechanisms under this setting [1]
- The authors do not compare to discrete diffusion LMs that support classifier guidance [1]


[1] Schiff, Y. "Simple Guidance Mechanisms for Discrete Diffusion Models." ICLR 2025.

---

> ### Author Rebuttal · Authors · 2025-07-31
>
> **Dear Reviewer X8kS,**
>
> Thank you for your thorough and insightful review. We value your feedback and respond to your comments as follows:
>
> ## **W1, Q1. Empirical Speedup**
>
> We appreciate your request for additional empirical results. To address this, we conducted a latency comparison across various methods using a consistent setup: sequence length of 64, batch size of 100, evaluated on a single A100 GPU. Latency values for DExperts, PPLM, LM-Steer, and DGLM are drawn from [1] and [2].
>
> As shown below, our proposed TTA method (with control) achieves markedly lower latency than prior classifier-guided diffusion methods such as SSD-LM and Diffusion-LM, and is competitive with autoregressive baselines in terms of runtime. Notably, while the original T = 1000 diffusion models (with or without control) incur substantial latency, our progressive timestep reduction allows us to match—or even surpass—the quality of these longer-run models with as few as 50 to 200 steps.
>
> | Method | Timestep | Speed ratio |
> | --- | --- | --- |
> | GPT-2 (base) | — | 1.0 |
> | DExperts | — | 2.6 |
> | PPLM | — | 270.1 |
> | LM-steer | — | 1.2 |
> | Diffusion-LM | 200 | 14.7 |
> | Diffusion-LM | 1000 | 72.7 |
> | SSD-LM | 1000 | 109.8 |
> | LD4LG | 250 | 2.2 |
> | DGLM | 50 | 4.4 |
> | TTA (w/o control) | 50 | 1.4 |
> | TTA (w/o control) | 1000 | 27.4 |
> | **TTA (with control)** | 50 | **2.0** |
> | TTA (with control) | 100 | 4.0 |
> | TTA (with control) | 200 | 7.9 |
> | TTA (with control) | 1000 | 39.2 |
>
> [1] Han et al., “Word Embeddings Are Steers for Language Models”, ACL 2024
>
> [2] Lovelace et al., “Diffusion Guided Language Modeling”, Findings of ACL 2024
>
> ## **W2. Comparison to discrete diffusion LM**
>
> Thank you for your thoughtful comment. We agree that discrete diffusion language models have shown strong performance in recent work, and it is important to consider them as baselines. In response to your suggestion, we evaluated our approach against the D-CBG framework for discrete diffusion guidance [1] on the sentiment control task.
>
> For this comparison, we used the publicly released **UDLM-LM1b** model and trained a Yelp sentiment classifier following their setup. We set the sampling steps to 128 and used the `siebert/sentiment-roberta-large-english` classifier for evaluation. The guidance strength parameter γ was swept from 1 to 50, and the best results are reported. Our method achieves higher sentiment accuracy and significantly lower perplexity compared to D-CBG, underscoring the effectiveness of our adaptive guidance.
>
> | **Method** | Timestep | Sentiment Accuracy | PPL |
> | --- | --- | --- | --- |
> | D-CBG (r=10) | 128 | 75.4 | 145.1 |
> | D-CBG (r=50) | 128 | 87.4 | 225.5 |
> | TTA (50, adaptive) | 100 | 91.0 | 25.4 |
>
> [1] Schiff et al., “Simple Guidance Mechanisms for Discrete Diffusion Models”, ICLR 2025.
>
> ## Q2. Applicability to Discrete Diffusion Frameworks
>
> Thank you for raising this important question. Indeed, our context-dependent token timestep allocation is compatible with discrete diffusion models, including the Discrete DIffusion classifier Guidance (**D-CBG)** framework [1].
>
> To demonstrate this, we integrated our adaptive allocation strategy into the **standard D-CBG framework**, evaluating it on the molecular property maximization task included in their original setup (QM9 dataset). Specifically, we focused on generating molecules that optimize **drug-likeness**, and compared performance against the baseline D-CBG model **without** timestep allocation under identical conditions. In our method, we adaptively modify the transition probabilities based on approximated classifier gradients, allowing the generation process to be guided more effectively.
>
> The table below shows the performance improvements from adaptive allocation across different guidance strengths (γ):
>
> | **γ** | **Method** | **Valid ()** | **Mean Property** |
> | --- | --- | --- | --- |
> | 1 | D-CBG | 989 | 0.474 |
> |  | + Adaptive Allocation | **998** | **0.494** |
> | 10 | D-CBG | 721 | 0.585 |
> |  | + Adaptive Allocation | **756** | **0.591** |
>
> Across guidance strengths, adaptive allocation improves both validity and target property scores, showcasing its value in discrete diffusion settings.
>
> We hope these additional results address your concerns. Thank you again for your thoughtful review and suggestions.

---

> > ### Author Response · Authors · 2025-08-05
> >
> > Dear Reviewer X8kS,
> >
> > Thank you again for your valuable feedback and for your careful consideration of our submission.
> >
> > As the author-reviewer discussion period comes to a close, we wanted to check in and see if there’s anything further we can clarify or expand upon. We’d be happy to respond to any remaining questions.
> >
> > Your comments helped guide several important revisions, and we’re grateful for your suggestions throughout the process.
> >
> > Warm regards,
> >
> > The Authors

---

> > ### Comment · Reviewer_X8kS · 2025-08-06
> > **Re: Rebuttal**
> >
> > I appreciate the authors' efforts in addressing my remaining concerns. It is especially insightful to see this additional analysis (in particular the extensions to discrete diffusion).
> >
> > Can the authors clarify why the speed ratio increases with higher timesteps? If the speed ratio is 1.0x for AR, shouldn't the speed ratio increase with lower timesteps? Or, if the reported metrics correspond to latency, can the authors clarify why GPT-2 achieves the lowest latency?

---

> > > ### Author Response · Authors · 2025-08-06
> > >
> > > Thank you for your thoughtful follow-up. We're pleased to hear that you found the additional analysis insightful.
> > >
> > > Regarding the speed ratio:
> > >
> > > Yes, the “speed ratio” reported in Table W1 reflects relative decoding latency, with GPT-2 (base) normalized to 1.0x. All other values indicate how much slower each method is compared to GPT-2.
> > >
> > > Unlike GPT-2, which performs unconditional generation, all other baselines—including diffusion models—introduce additional overhead due to their use of controllable generation. In our case, TTA without control (50 steps) is still 1.4× slower than GPT-2. We believe this difference is largely due to the absence of key–value caching in our diffusion-based implementation. While autoregressive models benefit significantly from caching and other inference-time optimizations, such techniques are not yet applied in our current setup. The reported values are based on a fixed sequence length of 64 for comparability; however, as the sequence length increases (e.g., to 256), the gap in decoding speed narrows due to the parallel nature of diffusion-based generation. In such settings, our method can even outperform GPT-2 in terms of decoding efficiency.
> > >
> > > We hope this explanation addresses your question and helps clarify the interpretation of the reported results.

---

### Official Review · Reviewer_sFKt · 2025-07-03

**Clarity:** 2
**Significance:** 3
**Originality:** 3
**Rating:** 3
**Confidence:** 4

**Summary:**

This paper proposes TTA-Diffusion, which is an inference time method to adaptively assign a differing number of diffusion time steps to every token in the sequence. The authors argue that this mechanism helps perform better classifier-guided controllable generation. The number of steps for a token is set by using signal from the classifier. The authors also show that their method is effective when diffusion models are progressively distilled to reduce the number of diffusion steps.

**Questions:**

Here are some questions I have about the paper:
- Does forgetting, as illustrated in figure 1, empirically occur? Doesn't the classifier guidance applied at every denoising step prevent "love" to change to "hate" when stearing for positive sentiment?
- The experimental setting for hypothesis 2 is unclear. Is classifier guidance being applied? If so, why does the probability drop?
- What is the input to the classifiers? Is it tokens, is it vectors? Appendix B2 mentions off-the-shelf classifiers are used for the different tasks, but these classifier operate on clean text. Are you using these same classifiers on noisy intermediate sequences during diffusion?
- Section 4.4 mentions for T=1000 the sentiment accuracy improves from 92.2 to 92.6, but these numbers are different from the ones reported in the table in the main paper and the appendix? Which table are these numbers from?
- Why are the results from TTA (5000) bad? Table 3a shows a very high perplexity for  TTA (5000). Table 9 also shows that TTA (5000) is worse than TTA (1000).
- In figure 2, how many samples correspond to each mean fluctuation ratio? Is the distribution close to uniform?

**Ethical Concerns:**

["NO or VERY MINOR ethics concerns only"]

**Final Justification:**

It is nice to see that the proposed method also woks in the discrete diffusion setting. However, my one remaining concern is that the linear schedule outperforms the adaptive schedule in many of the toxicity experiments. This makes me question the effectiveness of the proposed method.

**Limitations:**

Yes, the authors address limitations.

**Paper Formatting Concerns:**

No issues.

**Quality:**

3

**Strengths And Weaknesses:**

Strengths:
- TTA-Diffusion works with just 50-100 time steps. Compared to baselines, this should make TTA-Diffusion much faster.
- The analysis done for hypothesis 1 is a convincing motivating story for TTA-Diffusion.
- TTA-Diffusion is an inference-time method that can potentially be easily applied to many different diffusion methods.

Weaknesses:
- There are a few missing baselines. DGLM ([1]) is a very related method and there are no comparisons to DGLM. AR-Diffusion [2] proposes using something similar to linear schedule for assigning time steps, but there are also no comparisons to AR-Diffusion.
- The main contribution of the paper is to propose an adaptive strategy to assign time steps to each token. However, for their detoxification experiments they use a linear allocation schedule as mentioned in caption of table 1. Does this mean the adaptive strategy is only useful sometimes?
- The writing clarity can be improved. Some details are unclear; for e.g. is the time step allocation done once at the start of denoising or is the allocation changed at every step. Are the time steps for all tokens set according to the equation in 199 or only a subset of tokens?

[1] Lovelace, J., Kishore, V., Chen, Y., & Weinberger, K. (2024, August). Diffusion Guided Language Modeling. In Findings of the Association for Computational Linguistics ACL 2024 (pp. 14936-14952).
[2] Wu, T., Fan, Z., Liu, X., Zheng, H. T., Gong, Y., Jiao, J., ... & Chen, W. (2023). Ar-diffusion: Auto-regressive diffusion model for text generation. Advances in Neural Information Processing Systems, 36, 39957-39974.

---

> ### Author Rebuttal · Authors · 2025-07-31
>
> **Dear Reviewer sFKt,**
>
> Thank you for your thoughtful and constructive review. We address your comments below:
>
> ## **W1. Comparison with More Baselines**
>
> We appreciate your emphasis on broader baseline comparisons. Since the official implementation of DGLM [1] has not been released, we closely replicated their experimental setup and evaluated our results against the values reported in their paper.
>
> For sentiment and toxicity control tasks, we adopted the same evaluation datasets as DGLM: 5,000 OpenWebText prompts for sentiment and 5,000 neutral prompts from RealToxicityPrompts for toxicity. We report perplexity (measured by Olmo-1B) alongside control accuracy for a fair comparison. These results suggest that **TTA performs competitively with DGLM**, achieving lower perplexity and strong control performance.
>
> **Toxicity Control Results**
>
> | Method | Perplexity ↓ | Max Toxicity ↓ | Toxicity Prob. ↓ |
> | --- | --- | --- | --- |
> | DGLM (s=5) | 30.7 | 0.182 | 0.025 |
> | DGLM (s=20) | 38.2 | **0.101** | 0.005 |
> | TTA (50) | **29.2** | 0.119 | **0.004** |
>
> **Sentiment Control Results**
>
> | Method | Perplexity ↓ | Accuracy ↑ |
> | --- | --- | --- |
> | DGLM (s=100) | 28.6 | 96.6 |
> | DGLM (s=500) | **26.0** | 96.8 |
> | TTA (50) | 26.7 | **97.9** |
>
> Scaling AR-Diffusion [2] to 330M parameters on the C4 dataset was quite unstable. Instead, we adopted the AR-Diffusion objective (Multi-Level Diffusion) and fine-tuned our TTA(50) model accordingly. As shown below, this AR-style objective yields slightly better fluency, but at the cost of noticeably reduced classifier accuracy.
>
> **Sentiment Control Results**
>
> | Method | Perplexity ↓ | Accuracy ↑ |
> | --- | --- | --- |
> | TTA (50): AR-Diffusion | 38.7 | 77.4 |
> | TTA (50): adaptive | 42.0 | 85.6 |
>
> ## **W2. Adaptive allocation strategy in detoxification experiment**
>
> We appreciate your observation. Indeed, the detoxification experiments in Table 1 used a linear allocation schedule intentionally to demonstrate that various allocation strategies can be effectively applied within the same framework without requiring retraining.
>
> As the table below shows, both adaptive and linear schedules consistently outperform the default strategy across multiple timesteps. We chose to highlight results using the linear schedule to emphasize the flexibility and practicality of our method during inference.
>
> **Toxicity Control Results with different allocation methods**
>
> | **Timestep** | **Token Allocation** | **Avg. Toxicity** ↓ |
> | --- | --- | --- |
> | **50** | Adaptive (0.6) | 12.7 |
> |  | Linear | **12.5** |
> |  | Default | 14.0 |
> | **100** | Adaptive (0.6) | 12.3 |
> |  | Linear | **12.2** |
> |  | Default | 13.8 |
> | **200** | Adaptive (0.6) | **12.2** |
> |  | Linear | 12.2 |
> |  | Default | 13.6 |
>
> ## **W3. Experimental details on token update mechanism**
>
> Thank you for your suggestion. As mentioned in Line 161, timesteps are assigned to **all** tokens in the sequence, using the formula provided in Line 199. This allocation is dynamically updated at **each** denoising step based on the gradient magnitude with respect to the classifier. We have clarified this further in Section 3.2 of the revised manuscript.
>
> ## **Q1. Forgetting phenomena in Figure 1.**
>
> This is a great point. You're absolutely right that the classifier guidance step would eventually steer the token "hate" back toward something more positive. However, what we intended to illustrate in Figure 1 is that the *denoising step* of the diffusion mode (*before* applying classifier guidance) can sometimes undo or weaken the effect of the previous classifier update. This can lead to redundant or oscillating updates across timesteps, potentially causing instability in the generation process.
>
> The token transitions in Figure 1 (e.g., “clean → dirty”) are drawn from actual empirical traces. While drastic reversals (like “love → hate”) are rare, we frequently observe classifier-steered tokens reverting to a neutral or unintended form. Our proposed method is designed to mitigate this instability.
>
> ## **Q2. Experimental setting for hypothesis 2**
>
> Thank you for pointing this out. Yes, classifier guidance is **active throughout** the generation process in Hypothesis 2. Similar to Hypothesis 1, our goal in Hypothesis 2 is to measure how much the *diffusion denoising step* disrupts the classifier-guided updates from the previous step.
>
> Specifically, in Figure 3, the **"before diffusion step"** refers to the sequence *after classifier guidance is applied*, where key tokens are steered toward the target attribute. The **"after diffusion step"** then refers to the sequence *after the denoising step and before classifier guidance is re-applied at the next timestep*.
>
> The observed drop in classifier confidence reflects how much the denoising step alters or "forgets" the classifier's earlier updates—especially on key tokens with high gradient importance. A large confidence gap between these two stages indicates that the diffusion model is weakening the applied control signal, which motivates our analysis of update-forgetting.
>
> We have revised Section 2.2 to make this setup more explicit.
>
> ## **Q3. Details on external classifier application**
>
> Thank you for the question. While most off-the-shelf classifiers are designed to operate on clean tokenized text, we leverage the simplex-based representation during diffusion to obtain interpretable logit-based embeddings that can be directly fed into these external classifiers.
>
> This approach allows us to reuse standard classifiers without retraining them for the latent or embedding space. In fact, SSD-LM [1] demonstrated that applying external classifiers in the simplex domain is effective, even when the input is a noisy intermediate representation. By using this method, we avoid the need to train task-specific classifiers on the latent space, which is often unstable and computationally expensive.
>
> [1] Han et al., "SSD-LM: Semi-autoregressive Simplex-based Diffusion Language Model for Text Generation and Modular Control", ACL 2023
>
> ## **Q4. Clarification of Section 4.4**
>
> Thank you for pointing this out. We agree this section required clarification. The accuracy improvement from 92.2 to 92.6 corresponds to the **TTA (200)** variant evaluated with **T = 1000** inference steps. This result is reported in **Appendix Table 9**, where TTA (200) denotes the use of a reduced number of adaptive update steps (200), while the overall diffusion process still runs for 1000 steps.
>
> We have updated Section 4.4 to explicitly mention both **TTA (200)** and **T = 1000**, and to clearly reference Appendix Table 9 to avoid confusion.
>
> ## **Q5. Performance degradation of TTA (5000)**
>
> Thank you for this insightful question. We believe the performance drop of **TTA (5000)** is primarily due to a **mismatch between training and sampling conditions**. While TTA (1000) is trained with 1000 denoising steps and evaluated accordingly, TTA (5000) is trained with a much longer horizon of 5000 steps. When both models are sampled using the *same reduced number of inference steps* (e.g., under 1000), TTA (5000) tends to underperform due to its reliance on a longer denoising trajectory during training.
>
> This phenomenon is consistent with findings in other works—such as **SDTT** [1]—which show that a teacher model trained with a longer trajectory can actually perform worse than a distilled model when both are sampled under shorter inference budgets.
>
> [1] Deschenaux, Gulcehre, “Beyond Autoregression: Fast LLMs via Self-Distillation Through Time”, ICLR 2025.
>
> ## **Q6. Distribution of samples in Figure 2.**
>
> Thank you for bringing this up. The distribution of samples is not uniform across timesteps. The table below shows the distribution at two representative timesteps (**t = 899** and **t = 99**), where samples are separated into four equally ranged **fluctuation ratio bins**.
>
> At **higher timesteps** (e.g., t = 899), more samples fall into **higher fluctuation bins**, indicating greater variability early in the generation process. As the timestep **decreases** (e.g., t = 99), the distribution shifts, with more samples appearing in **lower fluctuation bins**—suggesting that fluctuation stabilizes over time. We’ve included a detailed figure of this distribution in **Appendix B**.
>
> **Percentage of samples in each fluctuation bin**
>
> | **Bin** | **t = 899** | **t = 99** |
> | --- | --- | --- |
> | Bin 1 (Lowest fluctuation) | 26.1% | 11.2% |
> | Bin 2 | 37.3% | 23.9% |
> | Bin 3 | 26.1% | 32.8% |
> | Bin 4 (Highest fluctuation) | 10.4% | 32.1% |
>
>
> Please let us know if any additional clarification would be helpful.
> Thank you again for your detailed and thoughtful feedback.

---

> > ### Author Response · Authors · 2025-08-05
> >
> > Dear Reviewer sFKt,
> >
> > We greatly appreciate your thoughtful and detailed review, as well as the depth with which you engaged with our work.
> >
> > With the discussion period nearing its conclusion, we wanted to reach out in case there are any additional points you'd like us to clarify. We would be glad to provide further explanation before the discussion period concludes.
> >
> > Thank you again for your constructive feedback, which has been instrumental in improving the clarity and rigor of our submission.
> >
> > Warm regards,
> >
> > The Authors

---

> > > ### Comment · Reviewer_sFKt · 2025-08-08
> > >
> > > Thank you for the reply. I have one more question. The linear schedule is similar to the schedule used in AR-Diffusion, right?

---

> > > > ### Author Response · Authors · 2025-08-08
> > > >
> > > > Thank you for your follow-up question.
> > > >
> > > > You're right — our linear schedule is conceptually aligned with the one used in AR-Diffusion, where the schedule is designed to support a left-to-right generation pattern. However, as noted in Section 3 of our paper, there is a key difference: while AR-Diffusion applies this scheduling during both training and inference, our approach applies it only at inference time. This allows our decoding method to remain independent of the training process, offering a more flexible and lightweight solution.
> > > >
> > > > We’ve also revised the manuscript to explicitly clarify that our linear schedule shares the same objective as that of AR-Diffusion.
> > > >
> > > > We hope this clarification is helpful, and please don’t hesitate to reach out if you have any further questions.

---

### Note · Authors · 2025-08-12

We sincerely thank the reviewers for their thoughtful reviews and constructive feedback. We are encouraged by their positive recognition of our work and appreciate the comments highlighting its strengths:

**Hypothesis & Analysis:** Clear identification of fluctuation-driven update-forgetting with well-motivated hypotheses and thorough empirical analysis (sFKt, X8kS, ThEE, NdQ8).

**Empirical Strength:** Consistent improvements in generation quality and speed, achieving strong performance even with a relatively small number of steps (sFKt, X8kS, NdQ8).

**Significance:** Addresses a timely challenge in diffusion generation with a practical, training-free approach (sFKt, ThEE).

---

Reviewers also raised important concerns, which we addressed as follows:

**Concern #1 — Applicability and completeness of baselines** (sFKt, X8kS, ThEE)

We added comparisons to DGLM, AR-Diffusion (sFKt 1), and discrete guidance (X8kS 2), where our method achieved higher control accuracy and lower perplexity. We also demonstrated transferability to both discrete (D-CBG) and continuous (Diffusion-LM) frameworks, confirming broad applicability.

**Concern #2 — Empirical validation: speed and scheduling** (X8kS, NdQ8, sFKt)

We added latency results showing TTA-Diffusion greatly reduces guidance overhead while maintaining competitive quality (X8kS 1, NdQ8 2). We showed adaptive outperforms other schedules for sentiment and matches linear for toxicity, both exceeding the default (sFKt 2).

**Concern #3 — Theoretical grounding and definition of update-forgetting** (ThEE)

We added semantics-aware metrics (BLEU, BERTScore) to complement strict token matching. We expanded the theoretical link between simplex and discrete diffusion and justified why adaptive allocation improves control accuracy.

---

We are pleased these additions resolved most concerns, with several reviewers acknowledging the strengthened analysis. In the revision, we will:

- Add full schedule comparisons to the Appendix.
- Integrate semantics-aware disruption metrics alongside token-level fluctuation.
- Expand the theoretical section to clarify links to discrete diffusion and perplexity bounds.

We are grateful for the reviewers’ recognition of this work’s significance. By introducing TTA-Diffusion and a principled framework to address update-forgetting, we aim to advance stable, controllable diffusion-based text generation.

Thank you for your time and consideration.

Sincerely,

The Authors

---

### Decision · Program_Chairs · 2025-09-17

**Decision:**

Accept (poster)

**Comment:**

I recommend accepting this paper, which addresses instability in classifier-guided diffusion language models. The authors identify "update-forgetting," where token-level updates fluctuate across timesteps, and propose TTA-Diffusion to dynamically allocate timesteps per token based on refinement needs. Experimental results show significant improvements in control accuracy and perplexity while reducing computational requirements. The reviewers acknowledged the clear problem formulation, thorough empirical validation, and practical inference-time approach. Although concerns were raised about novelty and broader applicability, the authors effectively addressed these through additional experiments demonstrating effectiveness across different diffusion frameworks. Despite being a borderline case, the paper's technical contribution and strong empirical results justify acceptance.